

# Retrieval of $O_2(^1\Sigma)$ and $O_2(^1\Delta)$ volume emission rates in the mesosphere and lower thermosphere using SCIAMACHY MLT limb scans

Amirmahdi Zarboo[1], Stefan Bender[1], John P. Burrows[2], Johannes Orphal[1], and Miriam Sinnhuber[1]

[1]Karlsruhe Institute of Technology, Germany
[2]University of Bremen, Germany

*Correspondence to:* Amirmahdi Zarboo (amirmahdi.zarboo@kit.edu)

**Abstract.** We present the retrieved volume emission rates (VER) from the airglow of both the daytime and twilight $O_2(^1\Sigma)$ band and $O_2(^1\Delta)$ band emissions in the mesosphere/lower thermosphere (MLT). The SCanning Imaging Absorption specintroMeter for Atmospheric CHartographY (SCIAMACHY) on-board the European Space Agency Envisat satellite observes upwelling radiances in limb viewing geometry during its special MLT mode over the range 50 to 150 km. In this study we use
the limb observations in the visible (595–811 nm) and near infrared (1200–1360 nm) bands.

We have investigated the daily mean latitudinal distributions and the time series of the retrieved VER in the altitude range from 53 to 149 km. The maximal observed VER of $O_2(^1\Delta)$ during daytime are typically 1 to 2 orders of magnitude larger than those of $O_2(^1\Sigma)$. The latter peaks at around 90 km, whereas the $O_2(^1\Delta)$ emissivity decreases with altitude, with the largest values at the lower edge of the observations (about 53 km). The VER values in the upper mesosphere (above 80 km)
are found to depend on the position of the sun, with pronounced high values occurring during summer for $O_2(^1\Delta)$. $O_2(^1\Sigma)$ shows secondary maxima during winter and spring, which are related to the downwelling of atomic oxygen after large sudden stratospheric warmings (SSW). Observations of $O_2(^1\Delta)$ and $O_2(^1\Sigma)$ airglow provide valuable information about both the chemistry and dynamics in the MLT and can be used to infer the amounts and distribution of ozone, solar heating rates and temperature in the MLT.

## 1  Introduction

The atmospheric airglow in the mesosphere and thermosphere above $\approx 60$ km is formed by fluorescent emission from excited states of atoms and molecules. Atoms and molecules in the mesosphere and lower thermosphere can be excited by absorption of solar radiation (photoluminescence), or by exothermic chemical reactions (chemiluminescence), see, e.g., Brasseur and Solomon (2006).

The processes that contribute to the airglow of atomic and molecular oxygen in the mesosphere and lower thermosphere are shown in Figure 1. The recombination of atomic oxygen (Barth, 1964), denoted as reaction (R1) and (k) in Figure 1, produces $O_2^*$ excited molecules:

$$O(^3P) + O(^3P) + M \longrightarrow O_2^* + M; \tag{R1}$$





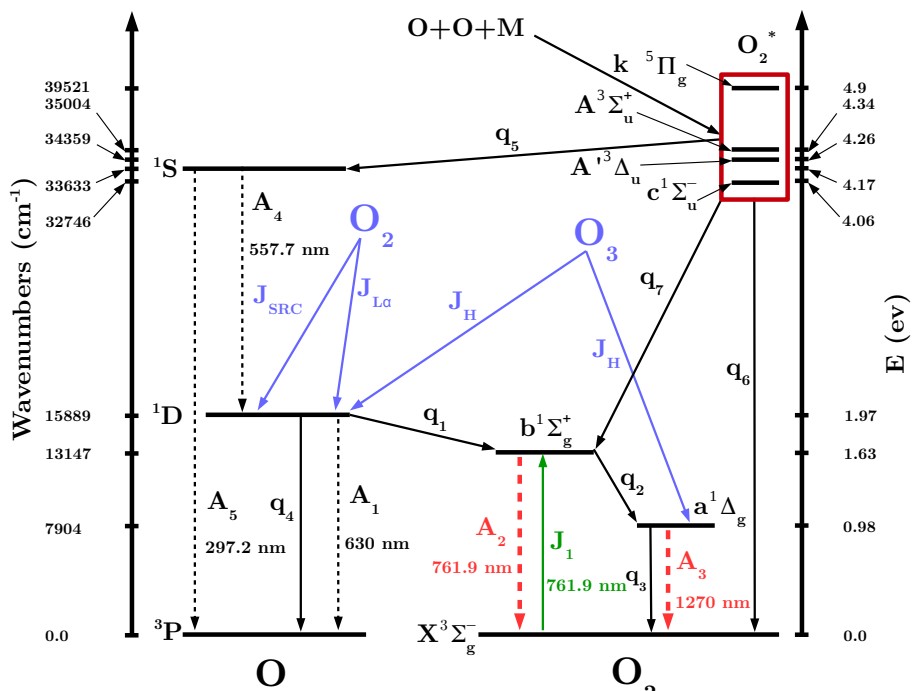

**Figure 1.** Schematic overview depicting processes that contribute to the production of Oxygen emission bands in the middle atmosphere. The black solid arrows are the most important gas phase reactions (k) and quenching reactions ($q_i$). The blue and green solid arrows are photo-dissociation and photo-excitation processes respectively ($J_i$). Dashed arrows correspond to spontaneous radiative emissions ($A_i$), and the red dashed arrows show the radiative emissions that are the subject of this study. (*) in $O_2^*$ denotes several excited states, see text for explanation.

The set of the electronic states of $O_2^*$, which are produced in reaction (R1), include the $c^1\Sigma_u^-$, $A'^3\Delta_u$, $A^3\Sigma_u^+$, or $^5\Pi_g$ states (Bates, 1995), and has been a matter of dispute for some years. The $c^1\Sigma_u^-$ state is considered the most probable (Slanger and Copeland, 2003).

The photolysis of ozone ($J_H$ in Figure 1, reaction (R2)) in the Hartley band ($\lambda < 310$ nm) leads to the first electronically excited state of atomic oxygen $O(^1D)$ and molecular oxygen $O_2(a^1\Delta_g)$ (DeMore, 1966):

$$O_3 + h\nu(\lambda \leq 310 \text{ nm}) \rightarrow O(^1D) + O_2(a^1\Delta_g). \tag{R2}$$

The photolysis of molecular oxygen in the Schumann-Runge continuum ($J_{SRC}$ in Figure 1, reaction (R3)) and at Lyman $\alpha$ ($J_{L\alpha}$ in Figure 1, reaction (R4)) leads to electronically excited oxygen atoms $O(^1D)$ (Nicolet, 1971):

$$O_2(X^3\Sigma_g^-) + h\nu(130 \leq \lambda \leq 175 \text{ nm}) \rightarrow O(^3P) + O(^1D), \tag{R3}$$



$$O_2(X^3\Sigma_g^-) + h\nu(\lambda=121.6 \text{ nm}) \rightarrow O(^3P) + O(^1D). \tag{R4}$$

Quenching (collisional de-excitation) processes are represented by black solid arrows and denoted by ($q_i$) in Figure 1. The $O_2^*$, produced by reaction (R1), can be quenched by atomic oxygen to produce $O(^1S)$ via reaction (R5) ($q_5$ in Figure 1) (Barth and Hildebrandt, 1961) or quenched by molecular oxygen to produce $O_2(b^1\Sigma_g^+)$ via reaction (R6) ($q_6$ in Figure 1) (Greer et al., 1981):

$$O_2^* + O(^3P) \rightarrow O(^1S) + O_2(X^3\Sigma_g^-), \quad (q_5) \tag{R5}$$

$$O_2^* + O_2(X^3\Sigma_g^-) \rightarrow O_2(b^1\Sigma_g^+) + O_2(X^3\Sigma_g^-). \quad (q_6) \tag{R6}$$

$O(^1D)$ by quenching with $O_2(X^3\Sigma_g^-)$, produces $O_2(b^1\Sigma_g^+)$ via reaction (R7) that combines $q_1$ and $q_4$ in Figure 1 (Mlynczak and Olander, 1995):

$$O(^1D) + O_2(X^3\Sigma_g^-) \rightarrow O_2(b^1\Sigma_g^+) + O(^3P). \tag{R7}$$

Photoabsorption of the solar radiation at 761.9 nm produces $O_2(b^1\Sigma_g^+)$ directly via

$$O_2(X^3\Sigma_g^-) + h\nu(\lambda=761.9 \text{ nm}) \rightarrow O_2(b^1\Sigma_g^+) \tag{R8}$$

in the sunlit mesosphere (Mlynczak et al., 1993), shown in Figure 1 as the radiative excitation $J_1$.

Then, according to reaction (R9) $O_2(b^1\Sigma_g^+)$ can be reduced in energy to $O_2(a^1\Delta_g)$ by collisions with any of the abundant species such as $O_2$, $N_2$, $CO_2$, O etc. (denoted by "M"), shown as $q_2$ in Figure 1 (Mlynczak and Olander, 1995):

$$O_2(b^1\Sigma_g^+) + M \rightarrow O_2(a^1\Delta_g) + M. \tag{R9}$$

Note that the above is an allowed (spin-conserved) process.

$O_2(a^1\Delta_g)$ can in turn be quenched via (R10) ($q_3$ in Figure 1) to produce $O_2(X^3\Sigma_g^-)$:

$$O_2(a^1\Delta_g) + M \rightarrow O_2(X^3\Sigma_g^-) + M, \tag{R10}$$

which is a spin forbidden process because the ground state is $O_2(X^3\Sigma_g^-)$.

Spontaneous radiative emissions are represented by dashed arrows and denoted by ($A_i$) in Figure 1. $O(^1S)$ decays to $O(^1D)$ by emitting 557.7 nm photons (the oxygen green line, reactions (R11), $A_4$ in Figure 1), which conserves spin and therefore the transition is allowed (Barth and Hildebrandt, 1961):

$$O(^1S) \rightarrow O(^1D) + h\nu(\lambda=557.7 \text{ nm}). \tag{R11}$$



Reactions (R1), (R5) and (R11) are commonly referred to as the Barth mechanism (Barth and Hildebrandt (1961); see, e.g., the review by Bates (1981)). The green line emission allows to deduce the atomic oxygen densities near 100 km, as shown for example by Lednyts'kyy et al. (2015). The oxygen 297.2 nm line is one of the prominent components of the ultraviolet nightglow (Slanger et al., 2006), and it is produced by $O(^1S)$ via reaction (R12) (Khomich et al., 2008):

$\quad$ $O(^1S) \rightarrow O(^3P) + h\nu(\lambda = 297.2\,\text{nm}),$ $\hfill$ (R12)

indicated by $A_5$ in Figure 1.

$\quad$ $O(^1D)$, produced from $((R2), J_H)$, $((R3), J_{SRC})$, $((R4), J_\alpha)$, or $((R11), A_4)$, can be deactivated to the ground state $O(^3P)$ by the spin forbidden emission:

$O(^1D) \rightarrow O(^3P) + h\nu(\lambda = 630.0\,\text{nm}),$ $\hfill$ (R13)

$\quad$ the 630.0 nm red line (Khomich et al., 2008) which is represented by $A_1$ in Figure 1.

$\quad$ Among the strongest features of the day and night airglow are the infrared atmospheric band $(a^1\Delta_g \rightarrow X^3\Sigma_g^-)$ and the atmospheric band $(b^1\Sigma_g^+ \rightarrow X^3\Sigma_g^-)$ of molecular oxygen (Wayne, 1994). These two spontaneous radiative emissions, which we deal with in this work, are represented by the thick red dashed arrows in Figure 1. They are emitted by the deactivation of the two excited states of the molecular oxygen $O_2(b^1\Sigma_g^+)$ at 761.9 nm via reaction (R14) ($A_2$ in Figure 1) and $O_2(a^1\Delta_g)$ at

$\quad$ 1.27 μm via reaction (R15) ($A_3$ in Figure 1) (Mlynczak et al., 1993):

$O_2(b^1\Sigma_g^+) \rightarrow O_2(X^3\Sigma_g^-) + h\nu(\lambda = 761.9\text{ nm}),$ $\hfill$ (R14)

$O_2(a^1\Delta_g) \rightarrow O_2(X^3\Sigma_g^-) + h\nu(\lambda = 1.27\ \mu\text{m}).$ $\hfill$ (R15)

$\quad$ Assuming that the processes in Figure 1 describe the photochemistry and chemistry, one can deduce ozone densities from
$\quad$ measurements of the infrared atmospheric volume emission rates in the $O_2(b^1\Sigma_g^+)$ and $O_2(a^1\Delta_g)$ bands (hereafter $O_2(^1\Sigma)$ and $O_2(^1\Delta)$ bands, respectively). For this, the rates of all of these processes such as $q_1$ and $q_4$ in Figure 1 and in the reactions (R7) and (R9) should be known (e.g., Evans et al. (1968); Thomas et al. (1983); Mlynczak and Olander (1995), Mlynczak et al. (2001)).

## 1.1 Previous measurements

$\quad$ The oxygen airglow was measured from space-borne platforms and rocket experiments in several previous studies. Measurements of the $O_2(^1\Sigma)$ band include the Fabry-Perot interferometer on the Dynamics Explorer 2 (DE-2) satellite (Skinner and Hays, 1985), that were used to study the overall brightness of the emission. The High-resolution Doppler imager (HRDI) on the Upper Atmosphere Research Satellite (UARS) (Hays et al., 1993) measured the Doppler shifts of rotational lines of the





$O_2(^1\Sigma)$ atmospheric band to determine the winds in the stratosphere, mesosphere, and lower thermosphere. The Wind Imaging Interferometer (WINDII) on the same satellite (Shepherd et al., 1993) measured wind, temperature, and emission rates. The TIMED Doppler Interferometer (TIDI) on the Thermosphere-Ionosphere-Mesosphere Energetics and Dynamics (TIMED) satellite (Killeen et al., 2006) performed remote sensing measurements of upper atmosphere winds and temperatures. The Re-

mote Atmospheric and Ionospheric Detection System (RAIDS) on the International Space Station's Kibo module (Christensen et al., 2012) measured the limb brightness of the (0,0), (0,1), and (1,1) vibrational band emissions from 80 to 180 km. The Optical Spectrograph and InfraRed Imaging System (OSIRIS), on board the Odin satellite (Sheese et al., 2010), was used to derive temperatures in the mesosphere-lower thermosphere region (MLT) from $O_2(^1\Sigma)$.

Previous measurements of the $O_2(^1\Delta)$ band include observations from the near-infrared spectrometer experiment on the

Solar Mesosphere Explorer satellite (SME). SME measured emission from $O_2(^1\Delta)$ produced by photolysis of $O_3$ (Thomas et al., 1984). The infrared atmospheric band airglow radiometer (IRA) aboard the satellite OHZORA measured the mesospheric ozone profile derived from $O_2(^1\Delta)$ emission (Yamamoto et al., 1988). One part of the Optical Spectrograph and InfraRed Imager System (OSIRIS) instrument on board the Odin satellite is a three channel infrared imager (IRI) that observes the scattered sunlight and the airglow from the oxygen infrared atmospheric band at 1.27 μm (Llewellyn et al., 2004). The TIMED

(Thermosphere Ionosphere Mesosphere Energetics and Dynamics)/SABER (Sounding of the Atmosphere using Broadband Emission Radiometry) data were used to measure the $O_2(^1\Delta)$ airglow emission by a channel with central wavelength of 1.27μm (Gao et al., 2011).

All of the above mentioned studies include satellite observations of only one of the $O_2$ bands, either $O_2(^1\Sigma)$ or $O_2(^1\Delta)$. Simultaneous measurements of both $O_2(^1\Delta)$ and $O_2(^1\Sigma)$ airglow were part of the Mesosphere-Thermosphere Emissions for

Ozone Remote Sensing (METEORS) sounding rocket experiment. It was launched from White Sands Missile Range, New Mexico (Mlynczak et al., 2001) and was used to derive ozone concentrations separately from each of the $O_2$ bands.

In this work, we retrieve volume emission rates (VERs) from the airglow of the $O_2(^1\Sigma)$ and $O_2(^1\Delta)$ bands in the meso-sphere/lower thermosphere (MLT) from the SCanning Imaging Absorption spectroMeter for Atmospheric CHartographY (SCIAMACHY Burrows et al. (1995), Bovensmann et al. (1999) and references therein) on board the European Space Agency

Envisat satellite. We present the retrieval algorithm and $O_2(^1\Sigma)$ and $O_2(^1\Delta)$ band volume emission rates. We analyze daily mean latitudinal distributions of VERs in the altitude range of approximately 50-150 km.

In section 2 we describe the SCIAMACHY dataset and our method to retrieve both the $O_2(^1\Delta)$ and the $O_2(^1\Sigma)$ volume emission rates. Results are presented in section 3, including the retrieved volume emission rates and first results on the temporal and spatial variations of the volume emission rates. We also include one example study of the relation between the

temporal variations and the sudden stratospheric warming in 2009. In section 4 we summarize the findings of our study and give conclusions.



## 2 Data and Methods

### 2.1 Data

The SCanning Imaging Absorption spectroMeter for Atmospheric CHartographY (SCIAMACHY) is a passive remote sensing spectrometer that observes back scattered, reflected, transmitted, or emitted radiation from the atmosphere and the Earth's surface in the 240-2380 nm wavelength range. The instrument is part of the atmospheric chemistry payload onboard the Envisat satellite, which was operational from March 2002 until April 2012. SCIAMACHY has three different viewing geometries: nadir, limb, and moon-sun occultations. From July 2008 until April 2012, SCIAMACHY observed the mesosphere and lower thermosphere region (MLT, 50–150 km) regularly twice a month. This special MLT limb mode scans the mesosphere and lower thermosphere in 30 limb points from 50 to 150 km altitude with a vertical spacing of about 3 km. These scans were scheduled in place of the nominal mode scans and there were 20 limb scans along one semi-orbit. Overall 84 days of mesosphere and lower thermosphere limb measurements were carried out. In this work, we use the visible and near infrared spectra from channel 4 (595–811 nm) and channel 6 (1200–1360 nm) in the MLT limb viewing geometry to retrieve volume emission rates (VERs) from the airglow of the $O_2(^1\Sigma)$ and $O_2(^1\Delta)$ bands.

To generate data for our study, we used the SCIAMACHY data set Level 1b version 8.02 and the SCIAMACHY command line tool `SciaL1c`[1] from the SCIAMACHY calibration tools. We selected two windows for each of the two bands: 750–780 nm for the $O_2(^1\Sigma)$ band (759–767 nm) and 1200-1360 nm for the $O_2(^1\Delta)$ band (1255–1285 nm). We subtract the spectrum measured at $\approx$ 360 km tangent height as a dark spectrum from the measured spectra at all of the other tangent heights. For the $O_2(^1\Delta)$ band, there are two masked points that appear in every scan located around 1262 nm and 1282 nm.

### 2.2 Daytime spectra

Daylight measurements by SCIAMACHY during the Envisat orbit begin with limb measurements of the twilit atmosphere (Bovensmann et al., 1999) which are located in the northern polar region (above 75°N). Our criterion to select daytime observations is that we require the tangent point solar zenith angle to be less than or equal to 88 degrees. Using this approach we avoid twilight measurements and all of the measurement points are located on the day-side.

Examples of the daytime calibrated spectra for orbit number 41455, measured on 03/02/2010 at a mean latitude of 17.3°N and a mean longitude of 94.3°E are shown in Figure 2a for the $O_2(^1\Sigma)$ band and in Figure 2b for the $O_2(^1\Delta)$ band.

The spectral region used to observe the daytime $O_2(^1\Sigma)$ spectrum includes a Rayleigh scattering background which perturbs the retrieval. Consequently it is necessary to estimate the Rayleigh scattering and subtract it from to yield the $O_2(^1\Sigma)$ emission spectra. This background scattering is attributed in part to the up-welling radiation, from multiple scattering in the lower atmosphere, and the terrestrial albedo. This results in an absorption signature for $O_2(^1\Sigma)$ (Sheese et al., 2010). To account for the multiple scattering and absorption by from the ground state $O_2(^3\Sigma_g)$ to the $O_2(^1\Sigma)$, a background signal comprising the

[1]https://earth.esa.int/web/guest/software-tools/content/-/article/scial1c-command-line-tool-4073



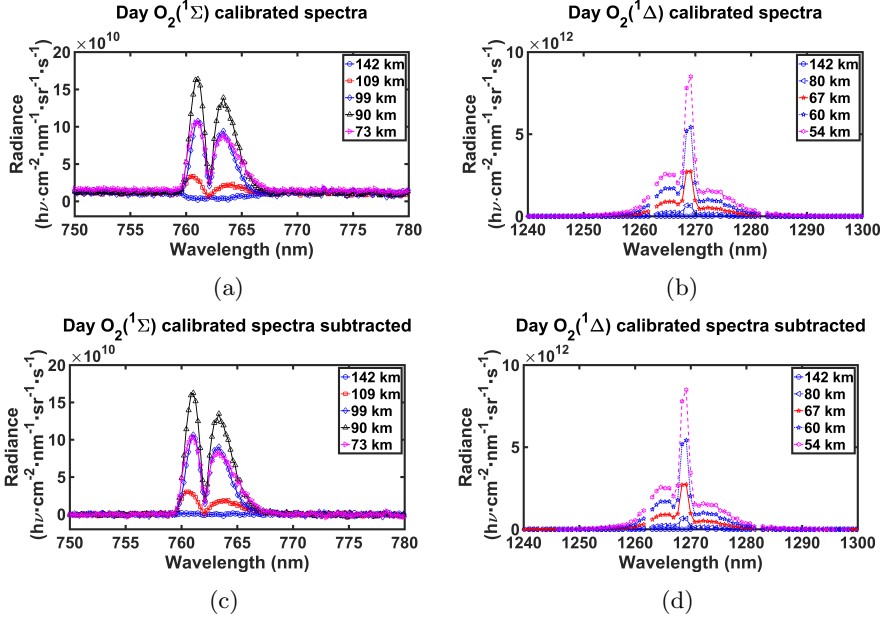

**Figure 2.** Examples of the daytime calibrated spectra, and the background-corrected spectra. (a) $O_2(^1\Sigma)$ on 03/02/2010, orbit 41455, mean latitude 17.3°N, mean longitude 94.3°E. (b) as (a) but for $O_2(^1\Delta)$ band. (c) as (a) but with background correction applied. (d) as (b) but with background correction applied.

$O_2(^1\Sigma)$ spectrum at the highest altitude ($\approx$ 148 km) scaled to the ratio of the mean of the out-of-band radiances is subtracted from the limb spectra at each tangent height. We consider the spectra in the 750–759 nm and 767–780 nm as out-of-band.

After this correction, we subtract a linear background from the whole signal in each level. An example of the background subtracted spectrum containing the $O_2(^1\Sigma)$ emission is shown in Figure 2c.

5    For the $O_2(^1\Delta)$ band, the absorption signature in the spectral background is negligible compared to the daytime $O_2(^1\Sigma)$ band spectra, therefore we only subtract a linear background from the observation. An example for the daytime $O_2(^1\Delta)$ band spectra with background subtracted is shown in Figure 2d.

## 2.3 Twilight spectra

Because the tangent point solar zenith angle for the sun being below the horizon varies with tangent altitude, we use equation (1)
10    to calculate the horizon angle for each tangent point ($R$ is the radius of the Earth and $h$ is the tangent point height):

$$\alpha_{\text{horizon}} = \frac{\pi}{2} + \cos^{-1}\left(\frac{R}{R+h}\right). \tag{1}$$





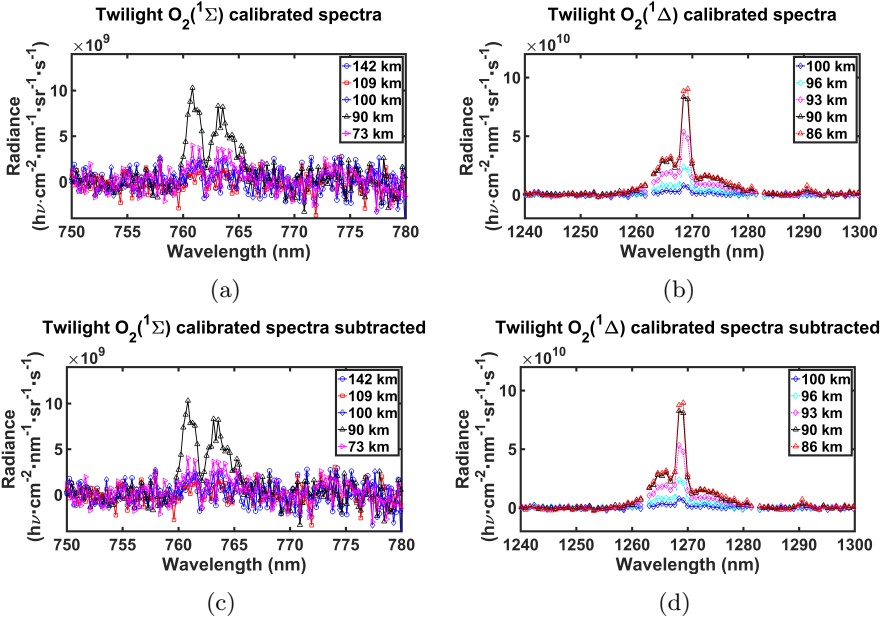

**Figure 3.** Examples of the twilight calibrated spectra, and the spectra from which the background $O_2(^3\Sigma_g)$ to $O_2(^1\Sigma)$ absorption has been subtracted. (a) for the 03/02/2010, orbit number 41455, mean latitude of 78.0°N, mean longitude of 226.5°E for $O_2(^1\Sigma)$ band. (b) as (a) but for the $O_2(^1\Delta)$ band. (c) as (a) but with background subtracted. (d) as (b) but with background subtracted.

As a criterion to select twilight data, we remove every limb scan in which at least one point measurement has a solar zenith angle less than given by equation (1). Based on this, we obtain the $O_2(^1\Sigma)$ and the $O_2(^1\Sigma)$ background subtracted twilight spectra shown in Figures 3a and 3c for the example orbit (41455). For the twilight $O_2(^1\Delta)$ band, we apply the same background subtraction as for daylight. Figure 3b shows the twilight $O_2(^1\Delta)$ spectra, and Figure 3d shows the background-corrected twilight $O_2(^1\Delta)$ spectra for the same example orbit (41455).

## 2.4 Retrieval

To invert the observed radiation to spectral emission rates, we set up 30 layers around the Earth, such that each layer is centered at one tangent height. We denote the observed spectral radiances by $\mathbf{Y}$, the path length of each of the observed line of sights through each of the atmospheric layers by $\mathbf{L}$, and the emission rate from the layer by $\mathbf{X}$. Assuming no self-absorption, this yields the linear relation (2):

$$\mathbf{Y} = \mathbf{LX} .$$ 
(2)





In equation (2), dim($\mathbf{Y}$) = number of atmospheric layers × number of spectral points = $30 \times 144$ for the $O_2(^1\Sigma)$ band and $30 \times 210$ for the $O_2(^1\Delta)$ band. Dim($\mathbf{L}$) = number of the atmospheric layers × number of tangent heights in each scan = $30 \times 30$. Dim($\mathbf{X}$) = number of atmospheric layers × number of spectral points in the corresponding wavelength interval = $30 \times 144$ for $O_2(^1\Sigma)$ and $30 \times 210$ for $O_2(^1\Delta)$.

We solve equation (2) by minimizing $||\mathbf{Y} - \mathbf{LX}||^2$ using standard least squares and normalization with the error covariance matrix $\mathbf{S}_y$, and obtain the inverted spectral emission intensity $\mathbf{X}$:

$$\mathbf{X} = (\mathbf{L}^T \mathbf{S}_y^{-1} \mathbf{L})^{-1} \mathbf{L}^T \mathbf{S}_y^{-1} \mathbf{Y} . \tag{3}$$

The error covariance matrix $\mathbf{S}_y$ has diagonal elements of the out-of-band variances of the background-corrected spectra in each altitude.

**3   Results**

**3.1   Inverted spectra**

Using our method in section 2.4, the emission intensities are calculated for the spectra described in sections 2.2 and 2.3. Examples of the emission intensity for daytime $O_2(^1\Sigma)$ are shown in Figure 4a and for daytime $O_2(^1\Delta)$ in Figure 4b.

    The spectral shape of the $O_2(^1\Sigma)$ band in the daytime $O_2(^1\Sigma)$ spectra, and of the $O_2(^1\Delta)$ band in the daytime $O_2(^1\Delta)$

spectra are clearly visible. We find that the largest values for the daytime $O_2(^1\Sigma)$ band are located at about 90 km altitude, and for the $O_2(^1\Delta)$ band at 54 km, noting that the SCIAMACHY MLT scan range is 50 to 150 km. We also note that the maximum $O_2(^1\Delta)$ emission intensities in each limb scan, are about two orders of magnitude larger than the maximum values of the $O_2(^1\Sigma)$ band in the corresponding limb scan.

    Figure 4c shows the twilight $O_2(^1\Sigma)$ emission intensity, and Figure 4d shows the twilight $O_2(^1\Delta)$ emission intensity for

the same orbit but retrieved from one of the three twilight MLT scans (see section 2.3). The $O_2(^1\Sigma)$ band is about one order of magnitude smaller than during daylight. The twilight $O_2(^1\Delta)$ band signal is more pronounced, but about two orders of magnitude smaller than during the day. The error bars in the panels of Figure 4 represent the square root of each of the diagonal elements of the retrieval error covariance matrix $\mathbf{S}_a$ for each altitude:

$$\mathbf{S}_a = \mathbf{G} \mathbf{S}_y \mathbf{G}^T \tag{4}$$

in which the contribution function matrix $\mathbf{G}$ is defined as:

$$\mathbf{G} = (\mathbf{L}^T \mathbf{S}_y^{-1} \mathbf{L})^{-1} \mathbf{L}^T \mathbf{S}_y^{-1} . \tag{5}$$

Figure 4a shows that the largest SNR of the emission intensities for daytime $O_2(^1\Sigma)$ are located around 83–99 km. The largest SNR are observed at the edges of the altitude range. Hereafter, we use the term 'significant' for data with a signal to noise ratio of greater than one.





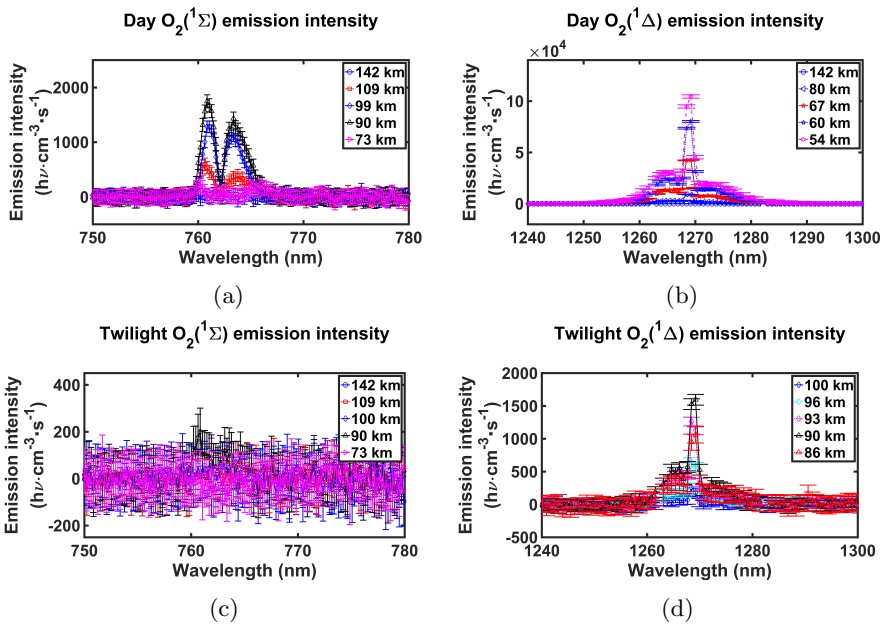

**Figure 4.** Examples of the emission intensities that are obtained by solving eq. (3). (a) for the daytime $O_2(^1\Sigma)$ band, on the date 03/02/2010, orbit number 41455, mean latitude of 17.3°N, mean longitude of 94.3°E. (b) as (a) but for $O_2(^1\Delta)$ band. (c) for the twilight $O_2(^1\Sigma)$ band on the date 03/02/2010, orbit number 41455, mean latitude 78.0°N, mean longitude 226.5°E. (d) as (c) but for $O_2(^1\Delta)$ band. Error bars represent the retrieval errors.

Figure 4c shows that the twilight $O_2(^1\Sigma)$ band emission intensities have large noise masking the signal. The daytime $O_2(^1\Delta)$ emission intensities are in general of higher SNR than the twilight $O_2(^1\Delta)$ emission intensities. The daytime $O_2(^1\Delta)$ emission intensities have the largest SNRs at the lowest altitudes (Figure 4b), whereas for the twilight $O_2(^1\Delta)$ the largest SNRs are located in the 83–96 km altitude range.

5    ## 3.2    Volume emission rates

We integrate the spectral emission intensity from 759 nm to 767 nm to obtain the $O_2(^1\Sigma)$ band integrated volume emission rate. Examples of the volume emission rate latitude-altitude distributions for one sample satellite orbit (41455 on 03/02/2010) for daytime $O_2(^1\Sigma)$ are shown in Figure 6a and for daytime $O_2(^1\Delta)$ in Figure 6b. The $O_2(^1\Sigma)$ VER has its maximum in the 90–98 km altitude range which is two orders of magnitude smaller than the $O_2(^1\Delta)$ maximum VER, as shown in Figure 5a.

10    Figure 5b shows that the $O_2(^1\Delta)$ maximum is located below 60 km, and sometimes shows secondary maxima in the range





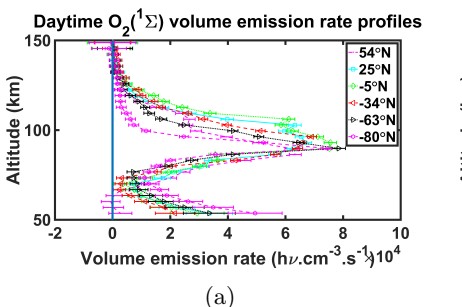
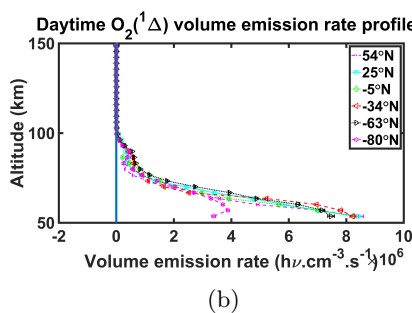

(a)              (b)

**Figure 5.** Typical profiles of the VER for different latitudes. (a) for daytime $O_2(^1\Sigma)$ VER, on the date 03/02/2010, orbit number 41455. (b) as (a) for daytime $O_2(^1\Delta)$ band. Error bars represent the retrieval errors.

80–90 km, which are at least one order of magnitude smaller than the primary maximum. This secondary maximum occurs especially around equinox times.

The measurement errors of the volume emission rates (not shown here) for different orbits show that the $O_2(^1\Sigma)$ volume emission rates are significant from 65 to 140 km, and do not depend on latitude. Inspecting the volume emission rate altitude
profiles (not shown here), we see that the $O_2(^1\Sigma)$ volume emission rates have largest SNR below 125 km and above 85 km. The best signal of the $O_2(^1\Delta)$ VER is below 95 km.

In a region above the South Atlantic and off the Brazilian coast, the Earth's magnetic field is anomalously low and the ionizing radiation can be increased by several orders of magnitude. This region is called the South Atlantic Anomaly (SAA, see for example Kurnosova et al. (1962)) and any spacecraft which crosses this region can give false instrument readings. In our
retrievals, the SNR of the volume emission rates in the orbits that cross this region are affected by the SAA. The most dramatic influence of the SAA on our dataset is on the $O_2(^1\Sigma)$ volume emission rates SNR, although the values are still significant in the 80-100 km altitude range. The SAA influences the $O_2(^1\Sigma)$ measurements more than the $O_2(^1\Delta)$ measurements.

### 3.3 Daily mean VER latitude-altitude distributions

We calculate the daily mean VERs as follows. We bin the measurements into 5° latitude bins. In each bin, the measurements
located within ±2.5° are attributed to that latitude, and averaged to daily mean VER. An example of the daily mean daytime $O_2(^1\Sigma)$ VER latitude-altitude distribution (on 03/02/2010) is shown in Figure 6c and of the daily mean daytime $O_2(^1\Delta)$ VER in Figure 6d (VERs with low signal to noise ratios and with large measurement errors are excluded). The daily mean $O_2(^1\Sigma)$ VERs have maximum values of about 1–2 orders of magnitude smaller than $O_2(^1\Delta)$. Similar to our results for a single orbit (section 3.2), we observe the largest $O_2(^1\Delta)$ VER below 60 km and the largest $O_2(^1\Sigma)$ VER at 90 km.
To assess the signal to noise ratio for the daytime VERs, Figure 6e shows the daily mean $O_2(^1\Sigma)$ VERs signal to noise ratios. $O_2(^1\Sigma)$ daytime VERs are significant in the 70–130 km altitude range. The twilight $O_2(^1\Sigma)$ VERs are significant between





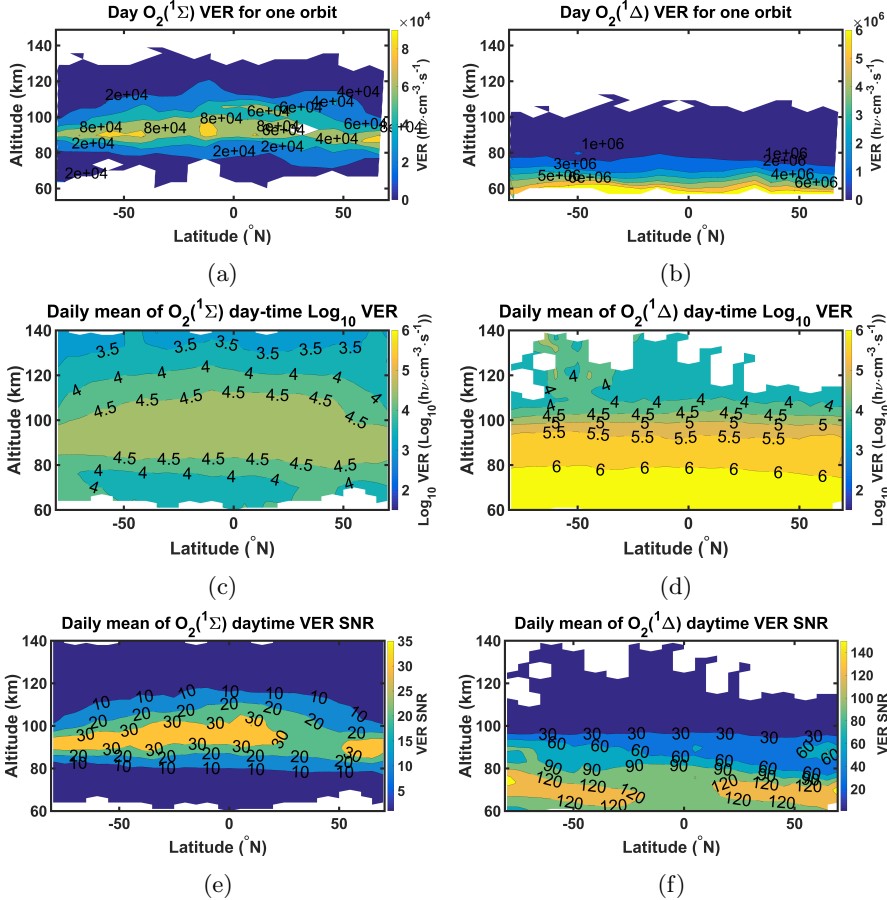

**Figure 6.** Latitude-altitude contours of the daytime VER. (a) for one satellite orbit, the $O_2(^1\Sigma)$ band, date: 2010/02/03, orbit number: 41455. (b) as (a) for $O_2(^1\Delta)$. Signal to noise ratios less than one and large noisy values are excluded. (c) and (d) as (a) and (b) respectively, averaged on the all of the orbits on the whole day of 03/02/2010, with the same logarithmic scale. Negative values are excluded. (e) Signal to noise ratios of the daily mean of $O_2(^1\Sigma)$ VERs. Areas where the signal to noise ratio is less than 1 are plotted in white. (f) as (e) for $O_2(^1\Delta)$ band.

84 km and 95 km around 80°N (not shown here). Figure 6f shows that the daytime $O_2(^1\Delta)$ VERs are significant below 105 km, the largest SNR are located around 70 km and north(south) of 20°N(20°S). The twilight $O_2(^1\Delta)$ VERs are significant in the altitude range of 83–97 km, around 80°N latitude (not shown).





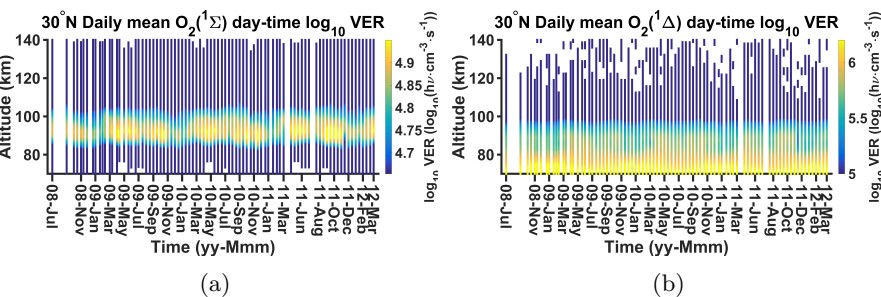

**Figure 7.** Time series of the daily mean VER. (a) for daytime $O_2(^1\Sigma)$ VER, 30°N, 07/2008 to 03/2012. (b) as (a) for daytime $O_2(^1\Delta)$ band.

## 3.4 Time series

In the following, we discuss the variation of the daily mean VERs versus latitude and time. First we will discuss the temporal variation in the mesosphere and lower thermosphere (70–140 km) at 30°N. Thereafter we will discuss the variation of the peak values, peak altitudes, and centroid altitudes as a function of time and latitude.

### 3.4.1 Time series at 30°N

By calculating the daily mean VERs for all of the days on which SCIAMACHY MLT limb scans are available, we obtain time series of the daily mean VERs from 07/2008 to 03/2012. An example of these time series of the daily mean daytime $O_2(^1\Sigma)$ VER, chosen for 30°N from all altitudes combinations is shown in Figure 7a, and for the daytime $O_2(^1\Delta)$ VER of the same location in Figure 7b.

We found a semi-annual variation with the strongest $O_2(^1\Sigma)$ signal in the 90-95 km range during May-June and September-November, and the lowest signal in December-March, with a secondary minimum in August. The highest values of the $O_2(^1\Delta)$ VER are located at the lowest altitude of observations, formed mostly by ozone photodissociation (Thomas et al., 1984). We observe secondary maximum values which mostly occur in May-June and September-November (approximately spring and autumn) in the 75–95 km altitude range. The secondary maximum of $O_2(^1\Delta)$ occurs in the same altitude range and with the same temporal variation as the $O_2(^1\Sigma)$ signal. This will be investigated in more detail in the following section.

### 3.4.2 Variation of peak values

Next we evaluate the variations of the maximal daily mean VERs in the mesosphere and lower thermosphere with respect to latitude and time. For this, we derive the maximal values between 85–100 km altitude from the daily mean VERs for both $O_2(^1\Sigma)$ and $O_2(^1\Delta)$, which are shown in Figures 8a and 8b. The peak altitudes are also obtained at the same time and will be discussed in section 3.4.3. The maxima of the $O_2(^1\Sigma)$ VER at middle to low latitudes (60°S–60°N) appear to be correlated with the maximum intensity of solar radiance. Additionally, we observe sometimes attenuations in the maximal values in the





late northern winters, mostly from late January until early March of each year. Also there are some high values at northern polar latitudes: in spring 2009, from autumn to spring 2009–2010, from autumn to winter 2010, and from autumn to spring 2011–2012.

The secondary maxima of the $O_2(^1\Delta)$ VER are also correlated with the maximum intensity of solar radiance. They are most

prominent in the hemispheric summer at high latitudes. The correlation of the VERs with the solar radiance suggests formation caused by solar light, either by ozone photolysis (R2) and (R7), or caused by larger abundances of atomic oxygen owing to stronger $O_2$ photolysis. In the second case, the formation of the excited states could also be due to the recombination of atomic oxygen (R1) and (R6). The maximal values at high latitudes, where the solar flux is low, in general suggest other sources such as recombination of atomic oxygen (Thomas et al., 1984). This is true in particular for $O_2(^1\Sigma)$, where high values occur at

high latitudes in winter and spring; these are probably caused by downward transport of thermospheric atomic oxygen into the mesosphere / lower thermosphere.

### 3.4.3 Variation of peak altitudes

The altitudes of the peak values of $O_2(^1\Sigma)$ and $O_2(^1\Delta)$ are shown in Figures 8c and 8d. The altitudes of the peak values of $O_2(^1\Sigma)$ roughly follows the maximum intensity of solar radiance, but shows highest values at low-to middle latitudes. The

low values of the maximum VERs, and the high altitudes of the maximal VERs at the outermost high latitudes in the northern winters, correspond to low signal to noise ratios and are below our significance level.

The highest peak altitudes of $O_2(^1\Delta)$ VERs occur in the hemispheric winter at high latitudes. The very high altitudes of the maximal VERs at the outermost high latitudes, which mostly occur in the hemispheric winter, correspond to low signal to noise ratios and are not statistically significant. The lowest peak altitudes (85 km, the lower edge) occur in summer at high

latitudes, where the highest peak values are also observed, see subsection 3.4.2 and in the sub-solar region.

### 3.4.4 Centroid altitudes

The estimation of the peak altitude is affected by instrument noise and vertical resolution. A more stable measure is the centroid altitude $h_{CA}$:

$$h_{CA} = \frac{\sum_i a_i v_i}{\sum_i v_i} \,, \tag{6}$$

where $v_i$ is the volume emission rate at altitude $a_i$. Figure 8e shows the centroid altitude for daytime $O_2(^1\Sigma)$ and Figure 8f for daytime $O_2(^1\Delta)$.

The maximum $h_{CA}$ of the $O_2(^1\Sigma)$ band are correlated with the (maximum intensity) of solar radiance. We also observe maximal values of $h_{CA}$ in a narrow band at northern polar latitudes, where the solar flux generally is low. This also suggests recombination of atomic oxygen as a source, as discussed in the subsection 3.4.2. The very low values of the $h_{CA}$s, at the

highest latitudes, mostly in the hemispheric wintertimes, correspond to low signal to noise ratios of the corresponding VERs (not shown here), and below our significance level.


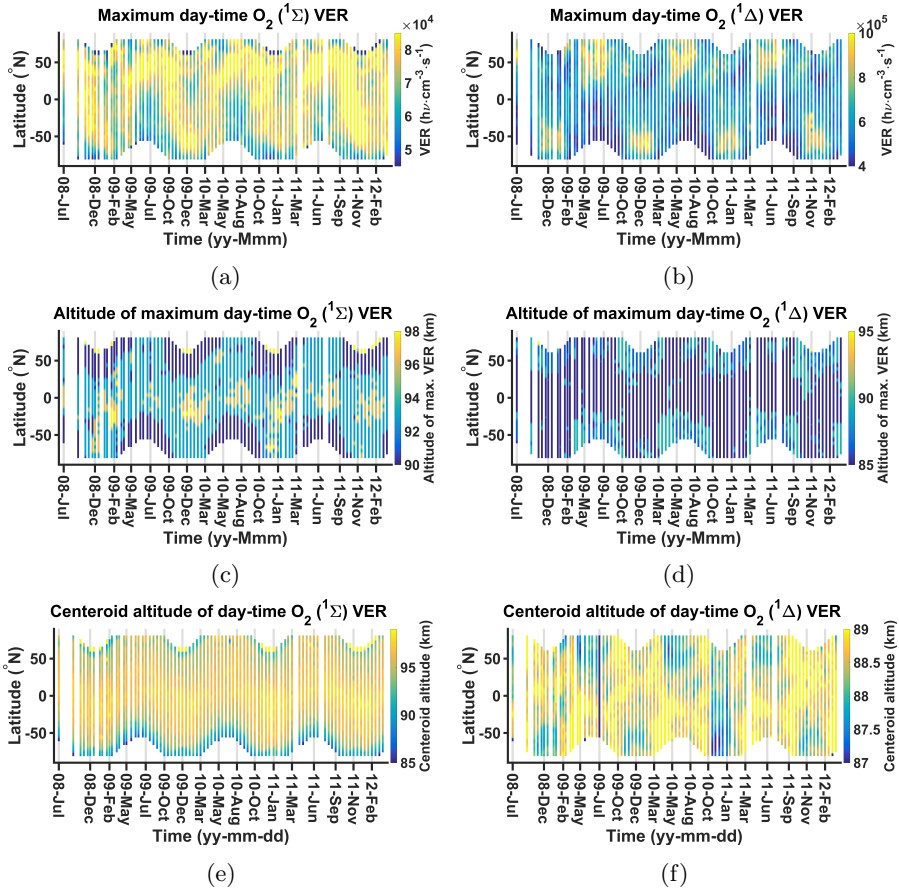

**Figure 8.** (a) Time series of the maximal daily mean $O_2(^1\Sigma)$ VER. (b) as (a) for $O_2(^1\Delta)$ band. (c) Time series of the altitudes of the maximum daily mean $O_2(^1\Sigma)$ VER. (d) as (c) for $O_2(^1\Delta)$ band. (e) Time series of the centroid altitudes of the daytime $O_2(^1\Sigma)$ daily mean VER (km). (f) as (e) for $O_2(^1\Delta)$ band.

The $h_{CA}$ for the $O_2(^1\Delta)$ band have low values coinciding with high values of the secondary maximum VERs. Large $h_{CA}$ for the $O_2(^1\Delta)$ band occur mostly in spring and autumn at almost all latitudes. Because the $O_2(^1\Delta)$ secondary maximums occur only in some places and times, $O_2(^1\Delta)$ $h_{CA}$ shows no clear pattern in Figure 8f. The very low values of the $h_{CA}$s, at the outermost high latitudes, mostly in the hemispheric wintertimes, correspond to low signal to noise ratios of the corresponding

5   VERs (not shown) and are below our significance level.





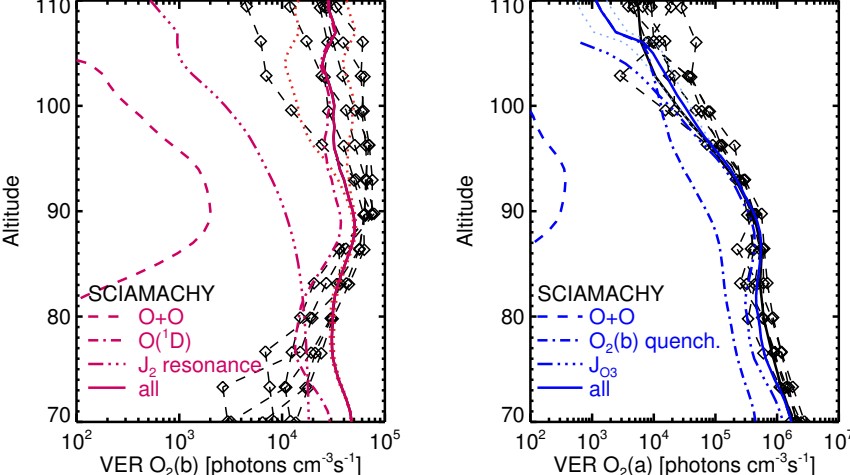

**Figure 9.** Modeled VERs of $O_2(^1\Sigma)$ (left) and $O_2(^1\Delta)$ (right). Solid lines are the mean of all model results for solar zenith angles 0, 10, 20, 30, 40, 50, 60, 70, 80, 88° considering all formation processes; dotted lines show the variability due to photolysis reactions ($\pm$ standard deviation). The dashed, dash-dotted, and dash-dot-dot lines show the contributions of individual formation processes: $O + O$, $O(^1D)$ quenching and resonant excitation for $O_2(^1\Sigma)$, $O + O$, $O_2(^1\Sigma)$ and ozone photolysis for $O_2(^1\Delta)$. Black lines with symbols are SCIAMACHY profiles at five latitudes observed on February 3, 2010 (see Figure 5).

Between November 2010 and February 2011 we observe an increase in the $O_2(^1\Sigma)$ $h_{CA}$ and a sudden drop in the $O_2(^1\Delta)$ $h_{CA}$, along with the attenuation of the secondary polar high values of the $O_2(^1\Sigma)$ $h_{CA}$. We suspect that these abrupt changes are related to a change in the altitude sequence of the satellite measurements during that time.

### 3.4.5 Discussion of temporal-spatial variation

The temporal-latitudinal variation in peak values and altitudes suggests that $O_2(^1\Sigma)$ at 85–100 km altitude is formed by a combination of ozone photolysis and atomic oxygen recombination. At high latitudes during winter and spring, atomic oxygen recombination dominates, but in the sub-solar region, ozone photolysis is more important. In contrast, the secondary peak of $O_2(^1\Delta)$ stems mainly from atomic oxygen recombination, in particular at high latitudes during winter.

The temporal-latitudinal variation in peak values and altitudes suggests that the daytime VERs of $O_2(^1\Delta)$ and $O_2(^1\Sigma)$ in the altitude range of 80–100 km are formed by a combination of ozone and $O_2$ photolysis and atomic oxygen recombination. At high latitudes during winter and spring, atomic oxygen recombination dominates, but in the sub-polar region, photolysis of ozone and $O_2$ is more important.

To test whether these conclusions are generally consistent with our understanding of the photochemical production and loss of $O_2(^1\Delta)$ and $O_2(^1\Sigma)$, a simple photochemical model covering the production and loss reactions summarized in Section 1 (R1 to R15) was set up. Photochemical equilibrium was considered for $O_2(c^1\Sigma_u^-)$, $O(^1S)$, $O(^1D)$, $O_2(^1\Delta)$, and $O_2(^1\Sigma)$.





Reaction rates were taken from the JPL recommendation (Burkholder et al., 2015) with the exception of the quenching of the intermediate $O_2(c^1\Sigma_u^-)$ state, which was taken from (Stegman and Murtagh, 1991) with $\delta$ coefficients from (Bates, 1988). Photolysis rates for solar zenith angles 0, 10, 20, 30, 40, 50, 60, 70, 80 and 88° were calculated using a fixed ozone profile using the 3dCTM model (Sinnhuber et al., 2012). The rate of photoexcitation of the ground state $O_2(X^3\Sigma_g^-)$ to the second

excited state $O_2(^1\Sigma)$ was calculated following (Bucholtz et al., 1986) but using recent line-strength provided by the HITRAN database at hitran.org (Rothman et al., 2013), yielding a rate of $2.04 \times 10^{-9}$ photons $cm^{-3}$ $s^{-1}$ above 70 km. Temperature, total air density, and the densities of $O_2$ and $N_2$ were taken from the NRLMSISE-00 model (Picone et al., 2002) at 10:00 local time at the equator. The production of $O(^1D)$ by photolysis of $H_2O$ and $CO_2$ in the Ly-$\alpha$ range was also considered, and $H_2O$ and $CO_2$ were assumed to have constant mixing ratios of 1 ppm ($H_2O$) and 380 ppm ($CO_2$). Ozone density was prescribed

adapted from a multi-year global average of SABER data in the altitude region 70–104 km (see Smith et al. (2013), Figure 3). Atomic oxygen was calculated from photochemical equilibrium of ozone considering only ozone photolysis and production by $O + O_2$. The resulting VERs of $O_2(^1\Delta)$ and $O_2(^1\Sigma)$ are shown in Figure 9, compared to the SCIAMACHY daytime data of February 3, 2010 shown in Figure 5). Considering that the ozone profile, $O_2$, $N_2$ and temperature are not chosen to fit those specific observations, the agreement is very good for both emissions, indicating that the main processes of $O_2(^1\Delta)$ and $O_2(^1\Sigma)$

formation and loss are reproduced well by this simple model. Also shown are the contributions of the individual production reactions: $O + O$, quenching of $O(^1D)$, and resonance excitation of $O_2(X^3\Sigma_g^-)$ for $O_2(^1\Sigma)$, $O + O$, quenching of $O(^1\Sigma)$, and ozone photolysis for $O_2(^1\Delta)$.

Below about 82 km, $O_2(^1\Sigma)$ is formed in about equal amounts by quenching of $O(^1D)$, while above, quenching of $O(^1D)$ dominates. The reaction of $O + O$ contributes about one order of magnitude less $O_2(^1\Sigma)$ than the other two branches even in

the region where it has the largest contribution (around 90 km). This is consistent with the ratio of emission intensities during twilight and during daytime of about a factor of ten as discussed in Section 3.1, considering that during night-time and twilight, $O_2(^1\Sigma)$ is formed solely by the $O + O$ reaction. The formation of $O_2(^1\Delta)$ is dominated by ozone photolysis at all altitudes, though below 90 km, $O_2(^1\Sigma)$ quenching contributes about 10–25%.

$O(^1D)$ is formed mainly by photolysis of $O_3$ in the below 90 km, by photolysis of $O_2$ above that altitude. During daytime,

the formation of both the $O_2(^1\Delta)$ and $O_2(^1\Sigma)$ states are formed photolysis in agreement with our observation that the peak maxima correlate with the maximum intensity of solar radiance. The contribution of the $O + O$ reaction generally is smaller by one to three orders of magnitude than the contribution of photolysis, in agreement with the lower VERs observed during twilight when photolysis is not a significant production process. However, it should be pointed out that the relation between the $O_2$ airglow and atomic oxygen by the $O + O$ reaction and de-excitation of the excited intermediate states such as $O_2(c^1\Sigma_u^-)$ is

probably quadratic to cubic; increased amounts of atomic oxygen, e.g. at high latitudes during winter when large amounts of atomic oxygen can be transported or mixed down from the lower thermosphere, can therefore increase the contribution of the $O + O$ reaction to the overall airglow considerably. This explains the observed enhancements of the airglow at high Northern latitudes in winter and spring, in particular as the centroid altitudes in these occasions range around 90 km, the altitude where the $O + O$ reaction has the strongest influence.



### 3.5 Relationship between the VER time series variations and sudden stratospheric warmings - example

Sudden Stratospheric Warmings (SSWs) are dynamical phenomena in the winter polar stratosphere caused by upward propagating planetary waves interacting with the mean flow (Matsuno, 1971). During the so-called recovery phase of SSWs, the reformation of the jet changes gravity wave propagation to the mesosphere. The induced change in the residual circulation results in an enhanced descent of air. This causes adiabatic warming and the stratopause reforms at altitudes as high as 75–80 km (Siskind et al., 2010). The unusual brightening of the OH airglow (Winick et al., 2009) presumably caused by enhanced downwelling of atomic oxygen. According to Harada et al. (2010), a major SSW event happened around 21 January 2009 (Figure 1 of that paper). We therefore expect to observe enhanced airglow at the end of January 2009.

Figures 10a and 10b show the time series from 12/2008 to 04/2009 of the daytime $O_2(^1\Sigma)$ and $O_2(^1\Delta)$ daily mean VER, averaged from 60°N to 70°N. One day after the initial day of the sudden stratospheric warming 2009 event, i.e., 22 January 2009, the daily zonal mean $O_2(^1\Sigma)$ and $O_2(^1\Delta)$ VERs show a reduced maximum intensity in the 82–87 km altitude range, indicating a decrease of atomic oxygen due to horizontal mixing and upwelling during the warming. We observe larger intensities about three weeks later, as expected from enhanced mixing with oxygen-rich thermospheric air after the SSW. After the recovery phase, the $O_2(^1\Delta)$ signal is less prominent compared to $O_2(^1\Sigma)$. A detailed analysis of this relationship is beyond the scope of this paper.

We also note that the signal to noise ratios of the VERs for $O_2(^1\Sigma)$ shown in Figure 10c is statistically significant. It is shown in Figure 10d that during and after the initial stage of the SSW event, the $O_2(^1\Delta)$ signal becomes weaker and stronger respectively, and the signal to noise ratio of the data are such that this behaviour is statistically significant.

### 4 Discussion and conclusions

We present the retrieval of daytime and twilight $O_2(^1\Sigma)$ and $O_2(^1\Delta)$ spectral emissions from MLT measurements of the airglow in limb viewing geometry from the SCIAMACHY instrument on-board Envisat. From the retrieved spectra, we calculate the band integrated VERs for both bands. The maxima of the $O_2(^1\Sigma)$ VER and the centroid altitude of the $O_2(^1\Sigma)$ are correlated with the maximum intensity of solar radiance. High values of maximum VER and centroid altitude are additionally seen at northern polar latitudes. The (30°N) time series of $O_2(^1\Sigma)$ VER shows a maximum in the 90–98 km altitude range. The maximum values correspond to high centroid altitudes for $O_2(^1\Sigma)$. The daily zonal (60°N–70°N) mean $O_2(^1\Sigma)$ VER show a reduced maximum intensity in the 82–87 km, in the initiation of the sudden stratospheric warming-2009 event, and an increase in intensity about three weeks later.

The maxima of the $O_2(^1\Delta)$ VER are also correlated with the maximum intensity of solar radiance, but they are most prominent in summer at high latitudes. The time series of $O_2(^1\Delta)$ VER is two orders of magnitude larger than $O_2(^1\Sigma)$ VER at its maximal values which are located below the observation altitude (<60 km), but it shows some secondary maxima with about one order of magnitude smaller than the primary maxima at 80-90 km. This happens especially in summer. The maximum VERs correspond to the low centroid altitudes for the $O_2(^1\Delta)$ band. The daily zonal (60°N–70°N) mean $O_2(^1\Delta)$ VERs show





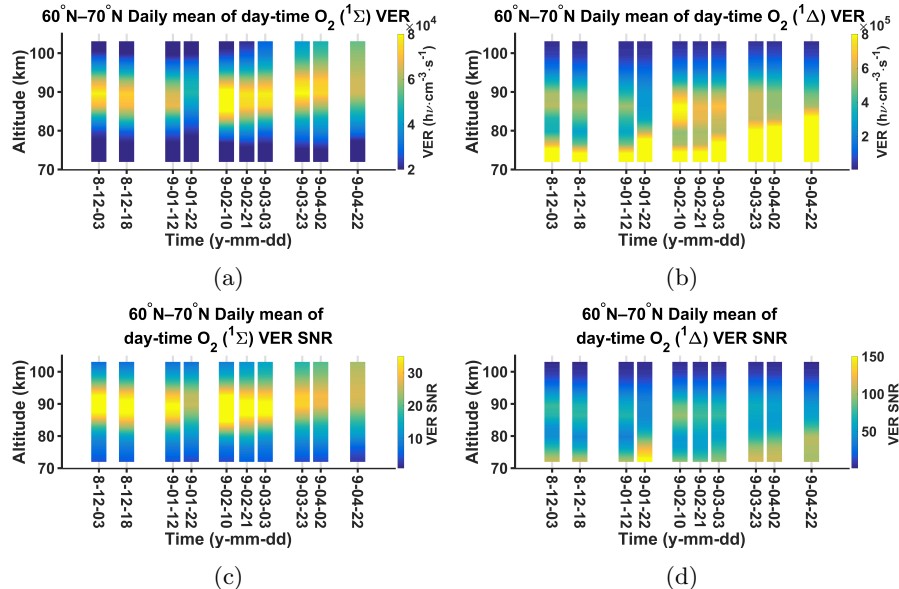

**Figure 10.** (a) Daily mean of the daytime $O_2(^1\Sigma)$ VER from date: 12/2008 to date: 04/2009 averaged between 60°N and 70°N latitude. (b) as (a) for $O_2(^1\Delta)$ band. (c) Daily mean of the signal to noise ratios of the daytime $O_2(^1\Sigma)$ VER from date: 12/2008 to date: 04/2009 averaged between 60°N and 70°N latitude. (d) as (c) for $O_2(^1\Delta)$ VER signal to noise ratios.

a reduced maximum intensity of 82–87 km, in the initiation of the sudden stratospheric warming-2009 event, and an increase in intensity about three weeks later, although the $O_2(^1\Delta)$ band signal is less prominent compared to $O_2(^1\Sigma)$.

The intensification of the VER during the sudden stratospheric warming in early 2009 presumably corresponds to the down-welling of the atomic oxygen following the warming, while the decrease is probably due to up-welling as well as horizontal mixing during the warming event.

Our results suggest that at low and middle latitudes $O_2(^1\Sigma)$ and $O_2(^1\Delta)$ abundances during daytime are dominated by photolysis of ozone below 90 km by photolysis of ozone and $O_2$ above 90 km as supported by our observed correlation with solar illumination, and consistent with the processes depicted in Figure 1. At high latitudes however, in particular during winter, atomic oxygen abundances might be a more important driver due to the recombination of $O + O$ and subsequent de-excitation via $O(^1S)$ and $O_2(^1\Sigma)$.

As the formation of the $O_2(^1\Delta)$ band is dominated by ozone photolysis, this band can be used to derive ozone densities directly. However, as quenching of $O_2(^1\Sigma)$ contributes about 10–25% to the overall production in 70–90 km, the accuracy of this retrieval can be improved considerably when both lines are available. The $O_2(^1\Sigma)$ band is dominated by quenching



of O($^1$D), and daytime O($^1$D) can be derived from observations of the O$_2$($^1\Sigma$) VER. However, below $\sim$90 km, the resonant excitation from the ground state has to be taken into account as well. The rate used here based on new line strength data from HITRAN is lower by about a factor of two compared to a similar estimate used for HRDI data as described in (Marsh et al., 2002), and lower by about a factor of four compared to the original estimate by (Bucholtz et al., 1986). We conclude that

5   the retrieval of O($^1$D) and ozone as the main source of O($^1$D) below 90 km from O$_2$($^1\Sigma$) is possible, but needs a careful estimation of the rate of resonant excitation.

*Acknowledgements.* Amirmahdi Zarboo and Miriam Sinnhuber gratefully acknowledge funding by BMBF grant 01LG1208A (ROMIC-MesoEnergy). The support of BMBF ROMIC project for the University of Bremen is also gratefully acknowledged. The SCIAMACHY project was funded by Germany with support from the Netherlands and Belgium as a national contribution to ESA Envisat.



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
