# Peer review of "Retrieval of $O_2(^1\Sigma)$ and $O_2(^1\Delta)$ volume emission rates in the mesosphere and lower thermosphere using SCIAMACHY MLT limb scans"

_Atmospheric Measurement Techniques, 2017_

## Referee Comment (RC1) · Anonymous Referee #1 · 23 Sep 2017

In this paper, the SCIAMACHY MLT limb scans are used to retrieve both O2(1$\sum$)andO2(1$\Delta$) dayglow emissions. The altitudinal and latitudinal distributions and the seasonal variations of both emissions are given and compared in brief. The possible photochemical processes contributed to the distributions are discussed. The responses of both emissions to the SSW in 2009 are analyzed simply. These new data are beneficial to the expansion of O2(1$\sum$)andO2(1$\Delta$) dayglow emissions database, and for that I commend the authors. However, some minor comments below should be addressed.

1.      Lines    10-12    on    page    1.        It    is    mentioned    that

"O2(1$\sum$)$shows secondary maxima during winter and spring, which are related to the downwelling of atomic oxygen after larges$

2. Line 2 on page 2. It is better to add a reference for the sentence "has been a matter of dispute for some years".

3. Line 4 on page 5. It is better to add "based on O2(1$\sum$)$emission" after "performed remote sensing measurements of upper atmosphere winds and temperature"$

4. Line 20-21. "Daylight measurements by SCIAMACHY during the Envisat orbit begin with limb measurements of the twilit atmosphere (Bovensmann et al., 1999) which are located in the northern polar region (above 75°N)." I can not catch the meaning of this sentence.

5. Lines 27-28 on page 9. Figure 4a just shows emission intensities at some altitudes; especially the intensity at 83 km is not shown in this figure. How to draw the conclusion from Figure 4a that the largest SNR of the emission intensities for daytime O2(1$\sum$)$are located around 83 - -99 km and the largest SNR are observed at the edges of the altitude range?$

6. Line 3 on page 10. "the largest SNRs at the lowest altitudes (Figure 4b),". Does the "lowest altitudes" means 54 km? If 54 km is the lowest altitude in the retrieval of O2(1$\Delta$) emission intensity in this paper? Is the intensity below 54 km not significant or can it not be observed?

7. Line 4 on page 10. Similar to question 5, the intensity at 83 km is not given in Figure 4d.

8. It would be better to add a simple introduction about Figure 5 before that about Figure 6, for example in line 7 on page 10.

9. Line 10 on page 10. Figure 5b gives the altitude distributions above 50 km (maybe 54 km). And some profiles have not reached their maxima at the lowest altitude given in the figure. The altitude distributions of O2(1$\Delta$) dayglow observed by the TIMED/SABER satellite often show its maximum around 50 km (usually lower than

50 km). The distributions of O2(1∆) dagylow were given by Mlynczak et al. [2007, JGR, 112, D15306]. It is suggested to compare the altitude distributions of O2(1∆) dayglow with that given by Mlynczak et al..

10. "around 80°N latitude" is mentioned in both lines 1 and 3 on page 12. Are both the VERs significant at only this latitude and not significant at other latitudes?

11. Lines 13-14 on page 14. "The altitudes of the peak values of $O2(1\sum) roughly follows the maximum intensity of solar radiance". This cannot be seen clearly from Figure 8c. What is the mean$

12. Lines 19-20 on page 14. The "maximal value" and "peak altitude" are derived from the profile between 85-100 km. Therefore, when the second peak does not appear or the second peak is not obvious, the derived maximal value must be the emission at 85 km and the peak altitude must be 85 km. In the situation, the peak altitude given here is not the altitude for a real peak. This means the peak values and peak altitudes at some latitudes or some time shown in Figure 8 are not for real peaks but for 85 km; however, they are for the real peaks when the second peaks are obvious. In fact, the lowest peak altitudes (85 km) occur not only in summer at high latitudes but also in the tropics in Figure 8d. It is better to solve this problem. At least, it should be explained in the paper in case some readers could think that all the maximum values and peak altitudes given in Figure 8 are for real second peaks.

13. Equation (6). Please give the altitude range.

14. Line 5 on Page 15. "hemispheric wintertimes" Please check if it should be "hemispheric summertimes".

15. Lines 2-3 on page 16. Please explain in brief why you suspect that these abrupt changes are related to a change in the altitude sequence of the satellite measurements during that time.

16. Lines 12-13 on page 18. The larger intensities could begin earlier (than three weeks later) as shown by Gao et al. [2011, JGR, 116, D19110]. You can compare the

results with those given in that paper.

17. Line 13 on page 18. It is better to add a citation for the sentence "as expected from enhanced mixing with oxygen-rich thermospheric air after the SSW".

18. Line 14 on page 18. It is mentioned that "After the recovery phase, the O2(1$\Delta$) signal is less prominent compared to O2(1$\sum$)". It is better if you can indicate the time when the recovery phase ended. In addition, it seems that the O2(1$\Delta$) signal is less prominent compared to O2(1$\sum$) after the recovery phase in Figure 10. However, the color bar in Figure 10a is different from that during the recovery phase; however it is still further stronger than the O2(1$\sum$) emission after the recovery phase. So is the SNR.

19. Line 2 on page 19. Similar to question 18.

20. Figure 5. Please revise the label in Figure 5.

21. Figure 6. Please explain in brief the blank regions in Figure 6 in the text.

22. Figure 7. Please revise the label (yy-Mmm) in Figure 7.

23. Figure 8. Please revise the labels (yy-Mmm and yy-mm-dd) in Figure 8. In addition, it is better to change the ranges of the color bars to show some characteristics described in the text more clearly; especially for Figure 8d, it is better to decrease the upper limit value of the color bar.

24. Labels in Figure 9. Add unit of "Altitude". It is better to use O2(1$\sum$) to replace O2(b) and use O2(1$\Delta$) to replace O2(a). It is better to use (photons cm-3 s-1) to replace [photons cm-3 s-1].

Please also note the supplement to this comment:
https://www.atmos-meas-tech-discuss.net/amt-2017-269/amt-2017-269-RC1-supplement.pdf

**Supplement:**

In this paper, the SCIAMACHY MLT limb scans are used to retrieve both $O_2(^1\Sigma)$ and $O_2(^1\Delta)$ dayglow emissions. The altitudinal and latitudinal distributions and the seasonal variations of both emissions are given and compared in brief. The possible photochemical processes contributed to the distributions are discussed. The responses of both emissions to the SSW in 2009 are analyzed simply. These new data are beneficial to the expansion of $O_2(^1\Sigma)$ and $O_2(^1\Delta)$ dayglow emissions database. The paper can be considered for publication in AMT. However, some minor comments below should be addressed.

1. Lines 10-12 on page 1.   It is mentioned that "$O_2(^1\Sigma)$ shows secondary maxima during winter and spring, which are related to the downwelling of atomic oxygen after large sudden stratospheric warmings (SSW).". What does the "secondary maxima" mean? It was not pointed out in the description in the text.

2. Line 2 on page 2. It is better to add a reference for the sentence "has been a matter of dispute for some years".

3. Line 4 on page 5. It is better to add "based on $O_2(^1\Sigma)$ emission" after "performed remote sensing measurements of upper atmosphere winds and temperatures".

4. Line 20-21. "Daylight measurements by SCIAMACHY during the Envisat orbit begin with limb measurements of the twilit atmosphere (Bovensmann et al., 1999) which are located in the northern polar region (above 75°N)." I can not catch the

meaning of this sentence.

5. Lines 27-28 on page 9. Figure 4a just shows emission intensities at some altitudes; especially the intensity at 83 km is not shown in this figure. How to draw the conclusion from Figure 4a that the largest SNR of the emission intensities for daytime $O_2(^1\Sigma)$ are located around 83–99 km and the largest SNR are observed at the edges of the altitude range?

6. Line 3 on page 10. "the largest SNRs at the lowest altitudes (Figure 4b),". Does the "lowest altitudes" means 54 km?  If 54 km is the lowest altitude in the retrieval of $O_2(^1\Delta)$ emission intensity in this paper?  Is the intensity below 54 km not significant or can it not be observed?

7. Line 4 on page 10. Similar to question 5, the intensity at 83 km is not given in Figure 4d.

8. It would be better to add a simple introduction about Figure 5 before that about Figure 6, for example in line 7 on page 10.

9. Line 10 on page 10. Figure 5b gives the altitude distributions above 50 km (maybe 54 km). And some profiles have not reached their maxima at the lowest altitude given in the figure. The altitude distributions of $O_2(^1\Delta)$ dayglow observed by the TIMED/SABER satellite often show its maximum around 50 km (usually lower than

50 km). The distributions of $O_2(^1\Delta)$ dagylow were given by Mlynczak et al. [2007, JGR, 112, D15306]. It is suggested to compare the altitude distributions of $O_2(^1\Delta)$ dayglow with that given by Mlynczak et al..

10. "around 80°N latitude" is mentioned in both lines 1 and 3 on page 12. Are both the VERs significant at only this latitude and not significant at other latitudes?

11. Lines 13-14 on page 14. "The altitudes of the peak values of $O_2(^1\sum)$ roughly follows the maximum intensity of solar radiance". This can not be seen clearly from Figure 8c. What is the meaning of the maximum intensity of solar radiance?

12. Lines 19-20 on page 14. The "maximal value" and "peak altitude" are derived from the profile between 85-100 km. Therefore, when the second peak does not appear or the second peak is not obvious, the derived maximal value must be the emission at 85 km and the peak altitude must be 85 km. In the situation, the peak altitude given here is not the altitude for a real peak. This means the peak values and peak altitudes at some latitudes or some time shown in Figure 8 are not for real peaks but for 85 km; however, they are for the real peaks when the second peaks are obvious.

In fact, the lowest peak altitudes (85 km) occur not only in summer at high latitudes but also in the tropics in Figure 8d.

It is better to solve this problem. At least, it should be explained in the paper in case some readers could think that all the maximum values and peak altitudes given in

Figure 8 are for real second peaks.

13. Equation (6). Please give the altitude range.

14. Line 5 on Page 15. "hemispheric wintertimes"

Please check if it should be "hemispheric summertimes".

15. Lines 2-3 on page 16. Please explain in brief why you suspect that these abrupt changes are related to a change in the altitude sequence of the satellite measurements during that time.

16. Lines 12-13 on page 18. The larger intensities could begin earlier (than three weeks later) as shown by Gao et al. [2011, JGR, 116, D19110]. You can compare the results with those given in that paper.

17. Line 13 on page 18. It is better to add a citation for the sentence "as expected from enhanced mixing with oxygen-rich thermospheric air after the SSW".

18. Line 14 on page 18. It is mentioned that "After the recovery phase, the $O_2(^1\Delta)$ signal is less prominent compared to $O_2(^1\Sigma)$". It is better if you can indicate the time when the recovery phase ended. In addition, it seems that the $O_2(^1\Delta)$ signal is less prominent compared to $O_2(^1\Sigma)$ after the recovery phase in Figure 10. However, the color bar in Figure 10a is different from that in Figure 10b. The $O_2(^1\Delta)$ emission after

the recovery phase is evidently weaker that that during the recovery phase; however it is still further stronger than the $O_2(^1\Sigma)$ emission after the recovery phase. So is the SNR.

19. Line 2 on page 19. Similar to question 18.

20. Figure 5. Please revise the label in Figure 5.

21. Figure 6. Please explain in brief the blank regions in Figure 6 in the text.

22. Figure 7. Please revise the label (yy-Mmm) in Figure 7.

23. Figure 8. Please revise the labels (yy-Mmm and yy-mm-dd) in Figure 8. In addition, it is better to change the ranges of the color bars to show some characteristics described in the text more clearly; especially for Figure 8d, it is better to decrease the upper limit value of the color bar.

24. Labels in Figure 9. Add unit of "Altitude". It is better to use $O_2(^1\Sigma)$ to replace $O_2$ (b) and use $O_2(^1\Delta)$ to replace $O_2(a)$. It is better to use (photons $cm^{-3}$ $s^{-1}$) to replace [photons $cm^{-3}$ $s^{-1}$].

---

## Referee Comment (RC2) · Anonymous Referee #2 · 19 Oct 2017

In the present manuscript, Zarboo et al. make an important contribution in presenting the volume emission rates of two molecular oxygen species as a function of altitude and season from limb-scans of the Earth's dayglow observed by the SCIAMACHY  instrument on the ESA Envisat satellite.  These data are particularly important in that the emissions of the two species, $O_2(^1\Sigma)$ and $O_2(^1\Delta)$, are measured simultaneously, allowing their variations to be correlated with photochemical processes in MLT region of the terrestrial atmosphere.  The authors develop a photochemical model for the atmospheric reactions involving $O_2(^1\Sigma)$ and $O_2(^1\Delta)$, use this model to draw conclusions from the emission measurements, and point to its potential use in monitoring ozone.

The present manuscript certainly has the potential for publication.  There is, however, one area of the manuscript that clearly requires correction, and several others, enumerated below by page and (line number(s)), that would benefit from revision.

**The portion of the manuscript requiring correction is:**

p. 2(1-3) and Figure 1  The recombination of atomic oxygen, reaction (R1) of the manuscript, is one of the primary reactions in the production of the emissions studied in the manuscript.  The authors list the likely product states of R1:   $^5\prod_g$ and the Herzberg states c $^1\sum_u^-$, A' $^3\Delta_u$, and A $^3\sum_u^+$, but then add "has been a matter of dispute for some years", without identifying what is disputed.  The statement is then made:  "The c $^1\sum_u^-$ state is considered the most probable (Slanger and Copeland, 2003)."

What the referenced Slanger and Copeland review actually says regarding Reaction R1 (their Eqn. 2):
"$O_2^*$ represents any of the seven states lying below the first dissociation limit, and it has been argued by Bates and others that the population distribution between these states can best be estimated statistically, in which case the  $^5\prod_g$ state should be generated in about 40% of all collisions."   Continuing in their Section 16 they state:
" It is now evident that much of the $O_2$ production derived from recombination is to be found in the A $^3\sum_u^+$ state; in a recent review it was concluded that 100% of the recombining atoms go through the Herzberg states."  Finally, the source of the authors' quenching rate constants (Stegman and Murtagh, 1991) "...set an upper limit of 10% on the production efficiency of  $O_2$ c $^1\sum_u^-$ ...".

Consequently, this reviewer finds little support for the authors' choice of the c $^1\sum_u^-$ state as the sole molecular product of R1.  Radiation from each of the Herzberg states is observed in the terrestrial nightglow (see Cosby et al., *J. Geophys. Res.*, 111, A12307, doi:10.1029/2006JA012023 and references therein) into *vibrationally excited* levels of the X $^3\sum_g^-$, a $^1\Delta_g$, and b $^1\sum_g^+$ states.  Collisional quenching of the Herzberg states yields *both* a $^1\Delta_g$ and b $^1\sum_g^+$ state products (Slanger and Copeland, 2003).

Admittedly, proper accounting for the products on R1 can be complex.   There is active research towards understanding this, e.g.  A. S. Kirillov, *Geomagnetism and Aeronomy*, 2012, Vol. 52, No. 2, pp. 242–247; *Chem. Phys. Letters* 592, 103-106 (2014).   Perhaps use of a surrogate state in the photochemical model will be appropriate.  However the authors should clearly delineate whatever approximations are being made.

**The following areas of the manuscript would benefit from revision:**

p. 3(19), p3(24),p. 3(22), p. 4(8):  "spin-conserved" and "spin forbidden".  The spin of the reactants and products is clearly defined in the associated equations and, indeed, most of the relevant transitions in both atomic and molecular oxygen are "spin-forbidden".  It might be better for the authors to also guide the reader towards the conclusion they intend to convey:  Is the particular reaction exceptionally fast? Is the particular reaction exceptionally slow or negligible?

p. 5(6)  Recommend changing "...(0,0), (0,1), and (1,1) vibrational band emissions..." to "... $O_2(^1\Sigma)$ (0,0), (0,1), and (1,1) vibrational band emissions...".  The way the paragraph begins on p. 4(25) makes it unclear that only $O_2(^1\Sigma)$ is being discussed.

p. 5(22-23)  A new section heading:  "1.2 Present Work" is needed here.

p. 6(11-12) "... we use ... channel 4 ... and ... channel 6 ...".  Using these two channels allows monitoring the 0-0 bands of $O_2(^1\Sigma)$ and $O_2(^1\Delta)$, but why not yet more? SCIAMACHY has eight channels covering uv to the far ir.  Could additional channels also have been used to monitor other oxygen emissions such as the green line or uv emissions from the Herzberg states?  It would useful to state the limitations or possibilities of the SCIAMACHY data set.

p. 6(16-17)  "We subtract the spectrum measured at $\sim$ 360 km tangent height as a dark spectrum from the measured spectra at all of the other tangent heights."  This is likely appropriate, but the reader is not shown the 360 km spectrum.  Is it intense?  Does it have features?  A bit of description regarding this dark spectrum would be helpful.

p. 8  Figure 3a and 3c.  These two figures show the twilight $O_2(^1\Sigma)$ radiance without- and with-background subtraction.  Yet to this reviewer, these figures appear identical.  If the background is truly negligible, it would be helpful to confirm that in the text at p. 8(3).

p. 16(3)  "... suspect that these abrupt changes are related to a change in the altitude sequence of the satellite measurements..."  This seems very important! Should not the effect of the altitude sequence on measured intensities be discussed a bit -

perhaps in Section 2 of the manuscript?  Should the data set presented in this manuscript be truncated at November 2010?

p. 17(1)  Are the rates A1, A3, A4 given in the "JPL Recommendation"?  If not, where are they coming from?

p. 17(2)  Is the "...quenching of the intermediate $O_2$ c $^1\Sigma_u^-$ state..." the $q_6$ and $q_7$ rates shown in Figure 1, but not otherwise mentioned in the manuscript?

p. 19(7)  Insert a comma (",") between "below 90 km" and "by photolysis of ozone"

---

## Author Comment (AC1) · 16 Nov 2017

[ACPD,manuscript]copernicus

amsmath

We would like to thank the referee for the careful review of the paper.

*1. Lines 10–12 on page 1. It is mentioned that "$O_2(^1\Sigma)$ shows secondary maxima during winter and spring, which are related to the downwelling of atomic oxygen after large*

[Figure]

*sudden stratospheric warmings (SSW)." What does the "secondary maxima" mean? It was not pointed out in the description in the text.*

Response: We mean "Additional polar high values of $O_2(^1\Sigma)$".

The manuscript is changed as follows:

"$O_2(^1\Sigma)$ emissions show additional high values at polar latitudes during winter and spring. These additional high values are presumably related to the downwelling of atomic oxygen after large sudden stratospheric warmings (SSW)."

*2. Line 2 on page 2. It is better to add a reference for the sentence "has been a matter of dispute for some years".*

Response: In our response we take into account the comments from both of the reviewers. The text now reads as follows:

$O_2^*$ represents any of the seven states below the first dissociation limit. Bates and others argue that the population distribution between these states can best be approximated statistically, in which the $^5\Pi_g$ state is produced in almost 40% of the collisions (Smith, 1984; Bates, 1992; Wraight, 1982). Most of the $O_2^*$ derived from recombination is found in the $A^3\Sigma_u^+$ state (Slanger and Copeland, 2003), and in a recent review Huestis concludes that all of the recombining atoms pass through the Herzberg states $c^1\Sigma_u^-$, $A'^3\Delta_u$, and $A^3\Sigma_u^+$ (Huestis, 2013). Stegman and Murtagh (1991) provide the quenching parameters resulted from analysing the measurements of the near-ultraviolet portion of the nightglow to fit the synthetic spectra of the Herzberg bands of $O_2$. These parameters set an upper limit of 10% production efficiency on the generation of $O_2(c^1\Sigma_u^-)$ in the atomic oxygen association reaction . Admittedly, proper accounting of the correct products of (R1) can be complex. Recent research has investigated this issue, e.g. Kirillov (2012, 2014). Therefore we assume

the production of a surrogate "hybrid" state $O_2^*$ in the photochemical model.

*3. Line 4 on page 5. It is better to add "based on $O_2(^1\Sigma)$ emission " after "performed remote sensing measurements of upper atmosphere winds and temperatures."*

This phrase was added to the corresponding place.

*4. Line 20–21. "Daylight measurements by SCIAMACHY during the Envisat orbit begin with limb measurements of the twilit atmosphere (Bovensmann et al., 1999) which are located in the northern polar region (above 75 °N)." I can not catch the meaning of this sentence.*

Response: The text was changed to make the meaning more clear. The manuscript is changed as follows:

"A typical orbit starts with a limb measurement of the twilit atmosphere, followed by the solar occultation measurement during sunrise over the North Pole and an optimized limb-nadir sequence. (Bovensmann et al., 1999). "

*5. Lines 27–28 on page 9. Figure 4a just shows emission intensities at some altitudes; especially the intensity at 83 km is not shown in this Figure. How to draw the conclusion from Figure 4a that the largest SNR of the emission intensities for daytime $O_2(^1\Sigma)$ are located around 83–99 km and the largest SNR are ovserved at the edges of the altitude range?*

Response: Indeed, this result corresponds to a draft version of the manuscript in which, the emission intensities have been shown for all of the measurement altitudes.

Whereas in this version, only some of the measurement altitudes are shown for the purpose of improving the visibility of the emission intensities inside the figure. Yes, it should be stated like this:

"Evaluating all altitudes, not shown here but indicated in Figure 4a, we observe the strongest signal of daytime $O_2(^1\Sigma)$ around 83–99 km."

*6. Line 2 on page 10. "The largest SNRs at the lowest altitudes (Figure 4b),". Does the "lowest altitudes" means 54 km? If 54 km is the lowest altitude in the retrieval of $O_2(^1\Delta)$ emission intensity in this paper? Is the intensity below 54 km not significant or can it not be observed?*

Response: In accordance with the answer to the previous question, not all of the measurement altitudes are included in Figure 4. Yes, the 54 km is the lowest observable altitude in our measurements. We changed the manuscript as follows:

"The daytime $O_2(^1\Delta)$ emission intensities are strongest at the lowest observable altitudes, i.e., 54 km (Figure 4b). The strongest twilight $O_2(^1\Delta)$ emissions are located in the 83–96 km altitude range (Figure 4d shows only a selection of altitudes)."

*7. Line 4 on page10. Similar to question 5, the intensity at 83 km is not given in Figure 4d*

Response: See reply to point #6 and the updated text therein.

*8. It would be better to add a simple introduction about Figure 5 before that about Figure 6, for example in line 7 on page 10.*

Thanks for pointing this out. A short introduction about Figure 5 has been added before the introduction about Figure 6. Change in manuscript: "Volume emission rate profiles for one sample satellite orbit (41455 on 03/02/2010) for daytime $O_2(^1\Sigma)$ and $O_2(^1\Delta)$ are shown in Figure 5a and Figure 5b respectively. Examples of the volume emission rate latitude-altitude distributions for the same orbit for daytime $O_2(^1\Sigma)$ are shown in Figure 6a and for daytime $O_2(^1\Delta)$ in Figure 6b."

*9. Line 10 on page 10. Figure 5b gives the altitude distributions above 50 km (may be 54 km). And some profiles have not reached their maxima at the lowest altitude given in the figure. The altitude distributions of $O_2(^1\Delta)$ dayglow observed by the TIMED/SABER satellite often show its maximum around 50 km (usually lower than 50 km). The distribution of $O_2(^1\Delta)$ dayglow were given by Mlynczak et al. [2007,JGR, 112, D15306]. It is suggested to compare the altitude distributions of $O_2(^1\Delta)$ dayglow with that given by Mlynczak et al.*

Response: An increase in the dayglow $O_2(^1\Delta)$ VER profiles is observed with decreasing altitude, but no turning point or maximum is observed.

We changed the manuscript as follows:

"The volume emission rate profile of dayglow $O_2(^1\Delta)$ observed by TIMED/SABER, often has its maximum around 50 km altitude, as shown for example in Figure 1 of Mlynczak et. al. (2007). Figure 5b shows that the SCIAMACHY MLT volume emission rate profiles are largest at the bottom of the observed altitude range, around 54 km. These VER profiles sometimes show secondary maxima in the range 80–90 km, which are at least one order of magnitude smaller than the largest SCIAMACHY VER. This secondary maximum occurs especially around equinox times."

[Figure]

10. *"around 80° N latitude" is mentioned in both lines 1 and 3 on page 12. Are both the VERs significant at only this latitude and not significant at other latitudes?*

Response: There are only twilight measurements at these latitudes. The significance relates to the altitude range. Therefore, since mentioning 80°N latitude, and 'significant data' is misleading, we removed these terms from this part of the text.

It should be noted that we make an improvement to the geometry of the retrieval and we get the smoother results with smaller variations. This is true for all of our results and we make the figures once again. So we get less noise and therefore the signal to noise ratio of the data is improved. As it can be seen, we have data with signal to noise ratio of greater than one in most of the altitudes and latitudes.

Change in the manuscript: "To assess the signal to noise ratio for the daytime VERs, Figure 6e shows the daily mean $O_2(^1\Sigma)$ VERs signal to noise ratios. We observe the strongest signal of daytime $O_2(^1\Sigma)$ in the 70–130 km altitude range. The strongest signal of the twilight $O_2(^1\Sigma)$ is observed between 84 km and 95 km (not shown here). Figure 6f shows that the stronger signal of daytime $O_2(^1\Delta)$ is observed below 105 km, with the strongest around 70 km. The largest signal of twilight $O_2(^1\Delta)$ is observed in the altitude range of 83–97 km (not shown). "

11. *Lines 13–14 on page 14. "The altitudes of the peak values of $O_2(^1\Sigma)$ roughly follows the maximum intensity of solar radiance". This can not be seen clearly from Figure 8c. What is the meaning of the maximum intensity of solar radiance?*

We refer to the light blue and yellow areas of the Figure 8c. We refer to the maximum intensity of solar radiance as the latitudes and times in which solar zenith angles have their lowest values in.

Change in the manuscript: "The altitudes of the peak values of $O_2(^1\Sigma)$ roughly follows

the maximum intensity of solar radiance, but shows highest values at low-to middle latitudes. We refer to the maximum intensity of solar radiance as the latitudes and times in which solar zenith angles have their lowest values in."

*12. Lines 19–20 on page 14. The "maximal value" and "peak altitude" are derived from the profile between 85–100 km. Therefore, when the second peak does not appear or the second peak is not obvious, the derived maximal value must be the emission at 85 km and the peak altitude must be 85 km. In the situation, the peak altitude given here is not the altitude for a real peak. This means the peak values and peak altitudes at some latitudes or some time shown in Figure 8 are not for real peaks but for 85 km; however, they are for the real peaks when the second peaks are obvious. In fact, the lowest peak altitudes (85 km) occur not only in summer at high latitudes but also in the tropics in Figure 8d. It is better to solve this problem. At least, it should be explained in the paper in case some readers could think that all the maximum values and peak altitudes given in Figure 8 are for real second peaks.*

Response: Indeed, with changing Figure 8b, 8d, and 8f such that only the regions where the secondary maxima of $O_2(^1\Delta)$ exist, we see that secondary maxima are confined to winter at mid-to high latitudes. That means they definitely do not follow the position of the sun - i.e. the highest photolysis rates of $O_3$. There are two possibilities for this - this is the region where the secondary ozone maximum is strongest, also atomic oxygen densities might be strongest due to enhanced mixing with the lower thermosphere.

Changes in the manuscript:

On page 14 line 11: " For this, we derive the maximum values from the daily mean VERs for $O_2(^1\Sigma)$ and between 85–100 km altitude for $O_2(^1\Delta)$, which are shown in Figures 8a and 8b respectively. Only those regions are shown in Figures 8b, 8d, and 8f where the secondary maxima of $O_2(^1\Delta)$ exist."

On page 14, line 19: " The secondary maxima of the $O_2(^1\Delta)$ VER are confined to winter at mid-to-high latitudes. "

On page 15, line 4: "There are two possibilities for secondary maxima of $O_2(^1\Delta)$. They happen in the region where the secondary ozone maximum is strongest. Also, atomic oxygen densities might be strongest due to enhanced mixing with the lower thermosphere. Detailed study of the processes which result in the formation of the secondary maxima of $O_2(^1\Delta)$ is beyond our work."

On page 15, line 13: "In the regions where the secondary maxima of $O_2(^1\Delta)$ happen, the peak altitudes occur in $\sim$84–89 km altitude range. "

On page 15, line 25: "The $O_2(^1\Delta)$ secondary maximums occur in winter at high latitudes. The values of the $O_2(^1\Delta)$ $h_{CA}range between \sim$88 km and $\sim$89 km altitude. "

On page 20, line 2: "The time series of $O_2(^1\Delta)$ VER is two orders of magnitude larger than $O_2(^1\Sigma)$ VER at its maximal values which are located below the observation altitude (<60 km), but it shows some secondary maxima with about one order of magnitude smaller than the primary maxima at 84-89 km. This happens in winter at high latitudes."

*13. Equation (6). Please give the altitude range.*

Response: It "ranges from $\sim$50 km to $\sim$150 km for $O_2(^1\Sigma)$ and from $\sim$85 km to $\sim$100 km for $O_2(^1\Delta)$."

The above is added to the text.

*14. Line 5 on Page 15. "hemispheric wintertimes" Please check if it should be "hemispheric summertimes".*

Response: This is the case only for $O_2(^1\Sigma)$ in Figure 8e, and is not the case for $O_2(^1\Delta)$ in Figure 8f. It is removed from the manuscript.

[Figure]

15. *Lines 2–3 on page 16. Please explain in brief why you suspect that these abrupt changes are related to a change in the altitude sequence of the satellite measurements during that time.*

Response: In our response we take into account the comments from both of the reviewers. The text now reads as follows:

"Figures 8c, 8d, and 8f, show a decrease in the altitude of the maximum $O_2(^1\Sigma)$, altitude of the maximum $O_2(^1\Delta)$, and $O_2(^1\Delta)$ $h_{CA}, respectively between November 2010 and February 2011. This is due to a change in the limb sequence so that tangent altitude su$

16. *Lines 12–13 on page 18. The larger intensities could begin earlier (than three weeks later) as shown by Gao et al. [20*

Response: We observe larger intensities on the next measurement day of SCIA-MACHY, i.e. 10 February 2009 which is about three weeks later than the previous measurement day of SCIAMACHY, i.e. 22 January 2009. To be more specific the following is inserted in the text:

Change in the manuscript: "Based on the temporal evolution of mesospheric temperature during the SSW event, Gao et al. (2011) divided the response in the mesosphere into three stages: prior to day 15 is considered the normal stage; days 15–22 correspond to the cooling stage; days post 22 correspond to the recovery stage. According to this, they reported that the $O_2$ nightglow brightness decreased by about a factor of 10 during the cooling stage and then increased by about a factor of 3 during the recovery stage relative to the normal stage. Figures 10a and 10b show the time series from 12/2008 to 04/2009 of the daytime $O_2(^1\Sigma)$ and $O_2(^1\Delta)$ daily mean VER, averaged from 60°N to 70°N. We observe that on the last day of the cooling stage, i.e., 22 January 2009, the daily zonal mean $O_2(^1\Sigma)$ and $O_2(^1\Delta)$ VERs show a reduced maximum intensity in the 82–87 km altitude range. We observe larger intensities on the next measurement day of SCIAMACHY about three weeks later, i.e., on 10 February 2009. This is expected from a decrease of atomic oxygen due to horizontal
mixing and upwelling during the cooling stage, and then downward extension of the MLT region with large mixing ratio of O during the recovery stage of the SSW (Gao et al., 2011)."

*17. Line 13 on page 18. It is better to add a citation for the sentence "as expected from enhanced mixing with oxygen–rich thermospheric air after the SSW".*

Response: The manuscript is changed as follows:

"This is expected from downward extension of the MLT region with large mixing ratio of O during the recovery stage of the SSW (Gao et al., 2011)."

*18. Line 14 on page 18. It is mentioned that "After the recovery phase, the $O_2(^1\Delta)$ signal is less prominent compared to $O_2(^1\Sigma)$". It is better of you can indicate the time when the recovery phase ended. In addition, it seems that the $O_2(^1\Delta)$ signal is less prominent compared to $O_2(^1\Sigma)$ after the recovery phase in Figure 10. However, the color bar in Figure 10a is different from that in Figure 10b. The $O_2(^1\Delta)$ emission after the recovery phase is evidently weaker than that during the recovery phase; however it is still further stronger than the $O_2(^1\Sigma)$ emission after the recovery phase. So is the SNR*

Response: The manuscript is changed as "On the measurement day of SCIAMACHY after the recovery phase, i.e. 23 March 2009, the relative difference in the $O_2(^1\Delta)$ signal is less prominent compared to the relative difference in $O_2(^1\Sigma)$."

*19. Line 2 on page 19. Similar to question 18.*

Response: The manuscript is changed to read as follows "although the relative difference in $O_2(^1\Delta)$ band signal is less prominent compared to the relative difference in $O_2(^1\Sigma)$."

*20. Figure 5. Please revise the label in Figure 5.*

Response: It is revised from "Volume emission rate ($h\nu.cm^{-3}.s^{-1}$)" to "Volume Emission Rate (photon.$cm^{-3}.s^{-1}$)"

*21. Figure 6. Please explain in brief the blank regions in Figure 6 in the text*

Response: The blank regions correspond to the regions with SNR less than 1.

Change in the manuscript: "The blank regions represent areas with signal to noise ratios of less than one." is added to page 10, lines 16–17.

*22. Figure 7. Please revise the label (yy-Mmm) in Figure 7.*

Response: It has been changed as suggested.

*23. Figure 8. Please revise the labels (yy-Mmm and yy-mm-dd) in Figure 8. In addition, it is better to change the ranges of the color bars to show some characteristics described in the text more clearly; especially for Figure 8d, it is better to decrease the upper limit value of the color bar.*

Response: It is done as suggested.

*24. Labels in Figure 9. Add unit of "Altitude". It is better to use $O_2(^1\Sigma)$ to replace $O_2(b)$ and use $O_2(^1\Delta)$ to replace $O_2(a)$. It is better to use (photons $cm^{-3}$ $s^{-1}$) to replace [photons $cm^{-3}$ $s^{-1}$].*

It is applied in the Figure 9.

**References**

Bates, D.: Nightglow emissions from oxygen in the lower thermosphere, Planetary and Space Science, 40, 211 – 221, doi:10.1016/0032-0633(92)90059-W, 1992.

Bovensmann, H., Burrows, J. P., Buchwitz, M., Frerick, J., Noël, S., Rozanov, V. V., Chance, K. V., and Goede, a. P. H.: SCIAMACHY: Mission Objectives and Measurement Modes, Journal of the Atmospheric Sciences, 56, 127–150, 1999.

Gao, H., Xu, J., Ward, W., and Smith, A. K.: Temporal evolution of nightglow emission responses to SSW events observed by TIMED/SABER, Journal of Geophysical Research: Atmospheres, 116, 2011.

Huestis, D. L.: Current Laboratory Experiments for Planetary Aeronomy, pp. 245–258, American Geophysical Union, doi:10.1029/130GM16, 2013.

Kirillov, A.: The calculation of quenching rate coefficients of O2 Herzberg states in collisions with CO2, CO, N2, O2 molecules, Chemical Physics Letters, 592, 103 – 108, doi:10.1016/j.cplett.2013.12.009, 2014.

Kirillov, A. S.: Model of vibrational level populations of Herzberg states of oxygen molecules at heights of the lower thermosphere and mesosphere, Geomagnetism and Aeronomy, 52, 242–247, doi:10.1134/S0016793212020077, 2012.

Slanger, T. G. and Copeland, R. A.: Energetic Oxygen in the Upper Atmosphere and the Laboratory, Chemical Reviews, 103, 4731–4765, doi:10.1021/cr0205311, 2003.

Smith, I. W. M.: The role of electronically excited states in recombination reactions, International Journal of Chemical Kinetics, 16, 423–443, doi:10.1002/kin.550160411, 1984.

Stegman, J. and Murtagh, D.: The molecular oxygen band systems in the UV nightglow: measured and modelled, Planetary and Space Science, 39, 595–609, 1991.

Wraight, P.: Association of atomic oxygen and airglow excitation mechanisms, Planetary and Space Science, 30, 251 – 259, doi:10.1016/0032-0633(82)90003-4, 1982.

---

## Author Comment (AC2) · 16 Nov 2017

[ACPD,manuscript]copernicus

amsmath

We would like to thank the referee for the careful review of the paper.

*p.2 (1-3) and Figure 1 The recombination of atomic oxygen, reaction (R1) of the manuscript, is one of the primary reactions in the production of the emissions stud-*

[Figure]

*ied in the manuscript. The authors list the likely product states of R1: $^5\Pi_g$ and the Hertzberg states $c^1\Sigma_u^-$, $A'^3\Delta_u$, and $A^3\Sigma_u^+$, but then add "has been a matter of dispute for some years" , without identifying what is disputed. The statement is then made: "The $c^1\Sigma_u^-$ state is considered the most probable (Slanger and Copeland, 2003)."*

Response: In our response we take into account the comments from both of the reviewers. The following has been added to the manuscript:

$O_2^*$ represents any of the seven states below the first dissociation limit. Bates and others argue that the population distribution between these states can best be approximated statistically, in which the $^5\Pi_g$ state is produced in almost 40% of the collisions (Smith, 1984; Bates, 1992; Wraight, 1982). Most of the $O_2^*$ derived from recombination is found in the $A^3\Sigma_u^+$ state (Slanger and Copeland, 2003), and in a recent review Huestis concludes that all of the recombining atoms pass through the Herzberg states $c^1\Sigma_u^-$, $A'^3\Delta_u$, and $A^3\Sigma_u^+$ (Huestis, 2013). Stegman and Murtagh (1991) provide the quenching parameters resulted from analysing the measurements of the near-ultraviolet portion of the nightglow to fit the synthetic spectra of the Herzberg bands of $O_2$. These parameters set an upper limit of 10% production efficiency on the generation of $O_2(c^1\Sigma_u^-)$ in the atomic oxygen association reaction . Admittedly, proper accounting of the correct products of (R1) can be complex. Recent research has investigated this issue, e.g. Kirillov (2012, 2014). Therefore we assume the production of a surrogate "hybrid" state $O_2^*$ in the photochemical model.

*p.3(19),p3(24),p.3(22),p.4(8): "spin–conserved" and "spin–forbidden". The spin of the reactants and products is clearly defined in the associated equations and, indeed, most of the relevant transitions in both atomic and molecular oxygen are "spin–forbidden". It might be better for the authors to also guide the reader towards the conclusion they intend to convey: Is the particular reaction exceptionally fast? Is the particular reaction*

*exceptionally slow or neglilgible?*

Done. The corresponding parts of the manuscript were modified.

*p.5(6) Recommend changing "...(0,0), (0,1), and (1,1) vibrational band emissions ..." to "...$O_2(^1\Sigma)$ (0,0), (0,1), and (1,1) vibrational band emissions...". The way the paragraph begins on p.4(25) makes it unclear that only $O_2(^1\Sigma)$ is being discussed.*

Changed as suggested.

*p.5(22-23) A new section heading: "1.2 Present Work" is needed.*

Done. A new section heading is added.

*p.6(11-12) "...we use ... channel 4 ... and ... channel 6 ...". Using these two channels allows monitoring the 0–0 bands of $O_2(^1\Sigma)$ and $O_2(^1\Delta)$, but why not yet more? SCIAMACHY has eight channels covering uv to the far ir. Could additional channels also have been used to monitor other oxygen emissions such as the green line or uv emissions from the Hertzberg states? It would be useful to state the limitations or possibilities of the SCIAMACHY data set.*

Response: Use of the additional channels covering the green line or UV is beyond our current work, and we refer to e.g. Lednyts'kyy et al. (2015).

Change in the manuscript:

"In this work, we use the visible and near infrared spectra from channel 4 (595–811 nm)

and channel 6 (1200–1360 nm) in the MLT limb viewing geometry to retrieve volume emission rates (VERs) from the airglow of the $O_2(^1\Sigma)$ and $O_2(^1\Delta)$ bands. Use of the additional channels covering the green line or UV is beyond our current work, and we refer to e.g. Lednyts'kyy et al. (2015). "

*p.6(16-17) "We subtract the spectrum measured at $\sim 360$ km tangent height as a dark spectrum from the measured spectra at all of the other tangent heights." This is likely appropriate, but the reader is not shown the 360 km spectrum. Is it intense? Does it have features? A bit of description regarding this dark spectrum would be helpful.*

Response: This spectrum contains some residual (read-out) patterns left from the calibration step and subtracting it from other spectra which have the same patterns cancels out that.

Change in the manuscript: " We subtract the spectrum measured at $\approx$ 360 km tangent height as a dark spectrum from the measured spectra at all of the other tangent heights. This spectrum contains some residual spectral (read-out) patterns left from the calibration step and subtracting it from other spectra which have almost the same patterns cancels out that.".

*p.8 Figure 3a and 3c. These two figures show the twilight $O_2(^1\Sigma)$ radiance without- and with-background subtraction. Yet to this reviewer, these figures appear identical. If the background is truly negligible, it would be helpful to confirm that in the text at p.8(3).*

Response: The following is added to the manuscript:

"It is apparent that the background signal is negligible for both of the $O_2(^1\Sigma)$ and $O_2(^1\Delta)$ twilight spectra."

*p.16(3) "...suspect that these abrupt changes are related to a change in the altitude sequence of the satellite measurements..." This seems very important!. Should not the effect of the altitude sequence on measured intensities be discussed a bit- perhaps in Section 2 of the manuscript? Should the data set presented in this manuscript be truncated at November 2010?*

Response: In our response we take into account the comments from both of the reviewers. The text now reads as follows:

"Figures 8c, 8d, and 8f, show a decrease in the altitude of the maximum $O_2(^1\Sigma)$, altitude of the maximum $O_2(^1\Delta)$, and $O_2(^1\Delta)$ $h_{CA}, respectively between November 2010 and February 2011. This is due to a change in the limb sequence so that tangent altitude su$

*p.17(1) Are the rates A1, A3, A4 given in the "JPL Recommendation"? If not, where are they coming from?*

Accepted. We used Einstein coefficients for the following transitons:

Change in the manuscript: "Einstein coefficients for the $O_2(^1\Sigma) \rightarrow O_2(X^3\Sigma_g^-)$ transition ($A_2$ in Figure 1) were taken from Mlynczak and Solomon (1993), for the $O(^1S) \rightarrow O(^1D)$ transition ($A_4$ in Figure 1) was taken from NIST atomic spectra database [1], for the $O(^1S) \rightarrow O(^3P)$ transition ($A_5$ in Figure 1) was taken from NIST atomic spectra database, and for the $O_2^*$ products (not given in Figure 1) were taken from Stegman and Murtagh (1991)."
* * *
[1]www.nist.gov

*p.17(2) Is the "...quenching of the intermediate $O_2c^1\Sigma_u^-$ ..." the q6 and q7 rates shown in Figure 1, but not otherwise mentioned in the manuscript?*

Accepted; Yes, that are the rates q6 and q7 given in Figure 1. Included a note about this in the text.

Change in the manuscript: "Reaction rates were taken from the JPL recommendation (Burkholder et al., 2015) with the exception of the quenching of the intermediate $O_2(c^1\Sigma_u^-)$ state (the q6 and q7 rates shown in Figure 1), which was taken from (Stegman and Murtagh, 1991) with $\delta$ coefficients from (Bates, 1988)."

*p.19(7) Insert a comma (",") between "below 90 km" and "by photolysis of ozone".*

Done.

**References**

Bates, D.: Excitation and quenching of the oxygen bands in the nightglow, Planetary and space science, 36, 875–881, 1988.

Bates, D.: Nightglow emissions from oxygen in the lower thermosphere, Planetary and Space Science, 40, 211 – 221, doi:10.1016/0032-0633(92)90059-W, 1992.

Burkholder, J., Sander, S., Abbatt, J., Barker, J., Huie, R., Kolb, C., Kurylo, M., Orkin, V., Wilmouth, D., and Wine, P.: Chemical Kinetics and Photochemical Data for Use in Atmospheric Studies–Evaluation Number 18, Nasa panel for data evaluation technical report, 2015.

Huestis, D. L.: Current Laboratory Experiments for Planetary Aeronomy, pp. 245–258, American Geophysical Union, doi:10.1029/130GM16, 2013.

Kirillov, A.: The calculation of quenching rate coefficients of O2 Herzberg states in collisions with CO2, CO, N2, O2 molecules, Chemical Physics Letters, 592, 103 – 108, doi:10.1016/j.cplett.2013.12.009, 2014.

Kirillov, A. S.: Model of vibrational level populations of Herzberg states of oxygen molecules at heights of the lower thermosphere and mesosphere, Geomagnetism and Aeronomy, 52, 242–247, doi:10.1134/S0016793212020077, 2012.

Lednyts'kyy, O., Von Savigny, C., Eichmann, K. U., and Mlynczak, M. G.: Atomic oxygen retrievals in the MLT region from SCIAMACHY nightglow limb measurements, Atmospheric Measurement Techniques, 8, 1021–1041, doi:10.5194/amt-8-1021-2015, 2015.

Mlynczak, M. G. and Solomon, S.: A detailed evaluation of the heating efficiency in the middle atmosphere, Journal of Geophysical Research: Atmospheres, 98, 10 517–10 541, 1993.

Slanger, T. G. and Copeland, R. A.: Energetic Oxygen in the Upper Atmosphere and the Laboratory, Chemical Reviews, 103, 4731–4765, doi:10.1021/cr0205311, 2003.

Smith, I. W. M.: The role of electronically excited states in recombination reactions, International Journal of Chemical Kinetics, 16, 423–443, doi:10.1002/kin.550160411, 1984.

Stegman, J. and Murtagh, D.: The molecular oxygen band systems in the UV nightglow: measured and modelled, Planetary and Space Science, 39, 595–609, 1991.

Wraight, P.: Association of atomic oxygen and airglow excitation mechanisms, Planetary and Space Science, 30, 251 – 259, doi:10.1016/0032-0633(82)90003-4, 1982.